# The Relationship between Diabetes Mellitus Type II and Intervertebral Disc Degeneration in Diabetic Rodent Models: A Systematic and Comprehensive Review

**DOI:** 10.3390/cells9102208

**Published:** 2020-09-29

**Authors:** Mohamed Mahmoud, Maria Kokozidou, Alexander Auffarth, Gundula Schulze-Tanzil

**Affiliations:** 1Department of Anatomy Paracelsus Medical University, Nuremberg and Salzburg, 90419 Nuremberg, Germany; mmagdibayoumi@gmail.com (M.M.); maria.kokozidou@pmu.ac.at (M.K.); 2Department of Orthopedics and Traumatology, Paracelsus Medical University, 5020 Salzburg, Austria; a.auffarth@salk.at

**Keywords:** intervertebral disc degeneration, diabetes mellitus type II, cytokines, glucose, culture, diabetic rat models, obesity, annulus fibrosus, nucleus pulposus, insulin, leptin

## Abstract

The number of diabetic patients grows constantly worldwide. Many patients suffer simultaneously from diabetes mellitus type 2 (T2DM) and intervertebral disc disease (IVDD), suggesting a strong link between T2DM and IVDD. T2DM rodent models provide versatile tools to study this interrelation. We hypothesized that the previously achieved studies in rodents approved it. Performing a search in the publicly available electronic databases according to our inclusion (e.g., experimental study with clearly outlined methods investigating IVDD in diabetic rodent models) and exclusion (e.g., non-experimental) criteria, we included 23 studies from 1992 to 2020 analyzing different aspects of IVDD in diabetic rodents, such as on pathogenesis (e.g., effects of hyperglycemia on IVD cells, sirtuin (SIRT)1/p53 axis in the interrelation between T2DM and IVDD), risk factors (e.g., high content of advanced glycation end-products (AGEs) in modern diets), therapeutical approaches (e.g., insulin-like growth factor (IGF-I)), and prophylaxis. Regarding their quality, 12 studies were classified as high, six as moderate, and five as low. One strong, 18 moderate, and three mild evidences of the link between DM and IVDD in rodents were found, while only one study has not approved this link. We concluded that T2DM has a devastating effect on IVD, particularly in advanced cases, which needs to be further evaluated.

## 1. Introduction

### 1.1. Basics

Low back pain (LBP) is one of the most common public health disorders in developed countries, with prevalence rates ranging from 12% to 35%. LBP affects up to 85% of people around the world and occurs at some point in their lives [1,2]. Degeneration of the intervertebral disc (IVDD) is considered to be an important underlying cause of LBP [3,4], back, neck, and radicular pains [5].

IVDs lay between the vertebrae in an alternating fashion giving to the spine its shape, flexibility, and high mobility in different axes and planes. IVDs represent up to 25% of the total spine length and act as distributors of loading and thereby resist compressive loads as well as tensile and shearing stresses [6,7]. The IVD is composed mainly of an external fibrous ring called annulus fibrosus (AF) and an internal jelly-like material called nucleus pulposus (NP), which in healthy discs appears like a whitish cotton-ball in healthy discs [8]. Both parts are connected by a transition zone. The NP has a hydrogel like consistency in healthy discs and becomes more fibrous in aged and degenerating IVDs [9]. AF attaches to the posterior longitudinal spinal ligament (PLSL) and fuses with the endplates of the upper and lower vertebrae. The AF consists of an external zone of concentric lamellar fibers of collagen type I and an internal zone of collagen fibers of type II. The localization of the NP within the IVD depends on loading conditions at the different segments (cervical, thoracic, or lumbar discs) [10,11]. Hence, differences among species exist as loading of the individual spine differs. In humans and rats, the NP in the lumbar spine is eccentrically and more dorsally localized between the central and posterior parts of IVD and exhibits properties of an incompressible water-filled cushion (Figure 1). The IVD contains glycosaminoglycans (GAGs) that comprise mainly chondroitin-4-sulfate, chondroitin-6-sulfate, and keratan-sulfate in young and aged people and dermatan-sulfate in the elderly [4,6,8,12,13,14]. In advanced aged humans, AF and NP begin to be dehydrated and partially collagenized [8]. However, in addition to aging, also metabolic diseases, such as diabetes mellitus, affect IVD homeostasis [5]. In this context, the studies of Fields et al. and Acevedo et al., showed that bone and IVD structure were indeed affected by type 2 diabetes mellitus (T2DM) in the University of California, Davis-Type 2 diabetes mellitus UCD T2DM rat model [15,16].

### 1.2. Diabetes Mellitus and Musculoskeletal Disturbances

DM represents a chronic degenerative disease, which results either from partial or complete insulin deficiency or cellular resistance against insulin receptors in target tissues [18,19]. DM is classified into four major types—T1DM, T2DM, gestational diabetes, and specific DM. T1DM and T2DM are the most common types and are discussed below, while specific or secondary diabetes and gestational diabetes are less common. Gestational diabetes mellitus (GDM) is defined as any glucose intolerance by the onset or during the course of pregnancy, regardless of whether it resolves or remains after delivery. Specific types of DM could be caused by genetic defects of β-cell function or insulin action or could be secondary to diseases of the exocrine pancreas, such as endocrinopathies, drug abuses, and/or to toxic effects of chemicals, infections, autoimmune disorders, or secondary to genetic syndromes associated with DM [20,21]. DM is classified into two main types: DM type I that is caused mainly by the reduction of insulin secretion from β-cells located in the Langerhans islets in the pancreas or from the destruction of those cells because of an autoimmune or non-autoimmune reaction, while DM type II results from a deficient or absent cellular response to insulin at the receptor level. DM type I represents about 10–15% of DM and occurs mainly in children, while DM type II represents 85–90% and is observed predominantly in adults and elderly people, especially in the case of obesity [19,22]. DM is an international major public health problem which is expected to affect 300 million people by 2025 worldwide [18,22,23] and more than 360 million by 2030 [22]. DM is characterized by wide disturbances in carbohydrate, lipid, and protein metabolic regulation, leading to an extensive and long-term dysfunction and various organ failures of the eyes, kidneys, liver, ears, cranial and peripheral nerves, heart and blood vessels, in addition to musculoskeletal tissues (bones and cartilages) [5,18,19,24]. In general, it mainly affects connective tissues [25].

DM is associated with many musculoskeletal disorders, such as Frozen shoulder, Dupuytren’s contracture, carpal tunnel syndrome, stiff hand syndrome, flexor tenosynovitis, joint stiffness, Charcot joint, gouty arthritis, osteoarthritis, rheumatoid arthritis, diabetic amyotrophy, diabetic muscle infarction, diffuse idiopathic skeletal hyperostosis, osteoporosis and osteoporosis-related fractures [22,25,26], and IVDD [5]. The underlying pathogenesis of joint stiffness is thought to be the increase of collagen cross-linking due to the non-enzymatic glycosylation of collagen with advanced glycation end-products (AGEs) formation [27]. However, AGEs were found in the NP of IVDs, mediating a dysregulation of aggrecan synthesis and IVD stiffening. Hence, their contribution to IVDD has been strongly suggested [3].

### 1.3. Diabetes Mellitus and Intervertebral Disc Degeneration

As many as 40% of cases of LBP go along with IVDD and upcoming evidence suggests that T2DM substantially contributes to the severity of IVDD [5]. The general pathogenesis of IVDD occurs by the loss and degradation of the disc’s extracellular matrix (ECM) components/molecules, such as proteoglycans, collagen, GAGs, and water content resulting from an imbalance of the catabolic and anabolic mechanisms, leading to wide histological and biochemical changes, such as increased levels of pro-inflammatory cytokines and associated enzymes [3,4,13]. Furthermore, the nutritional routes of the IVD and cellular viability and activities of the disc will be insulted [4,13]. Lifestyle factors, such as smoking and obesity, as well as background diseases, such as T2DM, contribute to IVDD. These effects are most likely mediated by malnutrition at the tissue level [28].

Krishnamoorthy et al. reported that the continuous accumulation of AGEs, which are associated with hyperglycemia in T2DM, was responsible for IVD stiffening and the following destructive cascade [3]. The T2DM-associated biochemical alterations lead to biomechanical and functional disturbances, followed by proceeding IVDD impairing in the whole spinal column as a functional unit. Hence, grading systems to assess T2DM-associated IVDD severity were developed. In 1990, Thompson et al. provided a macroscopical grading system [17]. More recently Raj [4] graded IVDD as follows: (Grade 0) normal unchanged nucleus; (Grade 1) annular tearing limited to the inner area of AF; (Grade 2) annular tears, destroying the entire disc architecture, but the outer contour of AF remains unaffected; (Grade 3) AF and PLSL are completely disrupted, including deformity of the entire disc [4]. A computer tomography (CT)-grading of IVDD was first stated in the 1990s, then initially modified by Aprill and Bogduk et al. in 1992 [29] based on MRI and finally modified in 1996 by Schellhas et al. [4,30,31]. Pfirrmann et al. developed in 2001 an MRI-based five-graded classification system of IVDD of the lumbar spinal discs, which comprises MRI signal intensity, disc structure, the distinction between AF and NP, and disc height on MRI scan [32]. Because of its high ambiguity, subjectivity, and non-applicability for young and old patients, the Pfirrman grading system was criticized by Xiong et al. [33], and by Griffith et al. [34], who provided/suggested a modification based on the widening of the grading of the IVDD from 5 to 8 categories and the increase of its power by including the elderly spines [33,34].

### 1.4. Rodents as Experimental Diabetic Models in IVDD

Since the pathogenesis of IVDD remains unclear, animal models are widely used as versatile tools to elucidate different aspects of IVD degeneration and to study its progression in detail. Jin et al. mentioned that the selection of the ideal animal model for the investigation of IVDD was more difficult than expected, and the following points should be considered: the complicated and partly unknown etiology and pathology of IVDD, the need of similarities of anatomical and pathological features of IVDD to those of humans, the reliability, the reproducibility, the labor efficiency, and, lastly, the costs invested in the experimental groups in regards to the size required to get statistically significant results [35].

A couple of detailed reviews regarding the used animal models in studying of IVDD degeneration and IVD regeneration (IVDR) exist already [35,36,37]. Such animal models used to study IVDD and IVDR were rodents, dogs, sheep, rabbits, goats, and primates [37]. Daly et al. mentioned numerous factors like the persistence of notochordal cells and the disc size, geometry, and mechanical forces that should be considered in the selection of the best-suited animal model to study IVDD and IVDR [36]. Rodents are considered to be a good choice model not only because of the low cost that allows statistically valid group sizes. Rodents could be a strong model, and the rat and mouse tail provide optimal access to IVDD, e.g., with the use of mechanical injury, asymmetrical compression, or administration of digestive enzymes. The advantages of rodent models are also the availability of numerous models of T2DM and obesity, the ease to manipulate them genetically, the short breeding span, the access to physiological and invasive testing, and the balanced cost-effectiveness [38,39]. In contrast, limitations include the fact that monogenic models are not representative of most human disorders [38]. In addition, there are other disadvantages, such as the persistence of notochordal cells in rodents, the obvious disc size discrepancy between humans and rodents (Figure 1), biomechanical differences in mechanical loading, and the ethical obstacles in the case of, e.g., creating bipedal mice [36] to mimic human conditions. During prenatal life, vertebrae and IVDs of all mammals arise from aggregation of the mesenchyme around the notochord and the following segmentation. In humans, the notochordal cells start diminishing after birth and completely disappear at adulthood, while in other species, including rats and mice, the notochordal cells distinctly remain until adulthood [35]. The persistence of notochordal cells limits the adequacy and relevance of implementation of rodents as the animal model of choice in the trials, which investigate cellular regeneration therapies of IVD because the resulting IVDR could not be correctly evaluated, whether it arises due to the applied therapies/agents or due to the potentially preexisting notochordal cells [36]. Genetically modified mice models are used to investigate the role of specific proteins in the etiology of IVDD pathogenesis [36]. Many methods are reported to create IVDD in rat or mouse tail. Such are genetic predisposition (through mutation creation) [35,37], mechanical loading (application of altered mechanical stresses by bending, cyclic chronic compression, and postural changes) [36,37], chemical digestion (through chemonucleolysis by using chemical agents or enzymes, which cause pathophysiological IVDD, such as chymopapain and chondroitinase ABC) [36,37], and physical or structural disruption (through surgical interferences as puncture and annulotomies) methods [35,36]. 

Numerous methods [40,41] on how to induce T2DM in the experimental rodent models have been reported, and the majority of them are mice models. Fajardo et al. proposed a novel classification system of the used diabetic rodent models of T2DM based on the following criteria: “(A) spontaneous or diet-induced, (B) mono- or polygenic etiology, (C) obese or lean body type, and (D) by the timing of T2D onset, either before or after skeletal maturity” [40].

T2DM rat models are monogenic obese models (Zucker diabetic fatty (ZDF) rats), polygenic obese models (Otsuka Long-Evans Tokushima fatty (OLETF) rat; Israeli sand rat (ISR)), induced obesity by high-fat diet (Nile grass rat and desert gerbil), and non-obese models (Goto-Kakizaki (GK) rat; spontaneous diabetic Torii (SDT) rat) [42]. The ISR [43] is an early onset obesity model used before and after skeletal maturity. It develops hyperinsulinemia and frank diabetes. No reports on its musculoskeletal disturbances were found. The ZDF rat is a model of moderate obesity and has a leptin receptor deficiency. Only the male individuals develop early hyperinsulinemia and later hyperglycemia, which is followed by decreased insulin. In comparison with controls, it shows decreased femoral length and diameter, in addition to bone formation in lumbar vertebrae. The GK rat is a spontaneous diabetic rat model that maintains glucose levels below 200 mg/dL. It has a shorter femur and smaller lumbar vertebral height, in addition to lower femoral and lumbar bone mass density (BMD) [42]. The spontaneous diabetic Torii (SDT) rat is a non-obese model of T2DM characterized by a blood glucose level of about 600 mg/dL at the age of 24 weeks and low insulin secretion caused by β-cell dysfunction. By advancing age, it exhibits decreased femoral and tibial BMD, reduced tibial three-point bending stiffness and peak load, and reduced bone formation rate [38,40]. The Zucker diabetic Sprague Dawley (SD) rat is a genetic-modified rat model through a leptin receptor mutation and develops rapidly a frank T2DM. It exhibits reduced femoral length and diameter, diaphyseal cortical thickness, and cortical volumetric BMD, biomechanical changes, and reduced vertebral BMD [38,40]. The induction of T2DM in SD rats using alloxan has the following drawbacks: high mortality rates, ketosis, and T2DM is reversible, while the use of streptozotocin (STZ), which is used mainly for the induction of T1DM, can also induce T2DM, having the following advantages: higher selectivity to β-cells, lower mortality rate, and T2DM is irreversible [44].

The OLETF rat is a polygenic rat model of moderate obesity that could be used in diabetic and prediabetic experiments. The musculoskeletal changes of the OLETF rat model are poorly reported [38,39,40,45]. In this context, the studies of Fields et al. and Acevedo et al. showed that bone and IVD structure was indeed affected by T2DM in the UCD T2DM rat model [15,16].

T2DM mice models are monogenic obese models (leptin receptor-deficient diabetic and obese (Leprdb/db and Lepob/ob) mice), polygenic obese models (Tsumura Suzuki obese diabetes (TSOD); Nagoya-Shibata-Yasuda (NSY); M16; C57Bl/6J; TallyHo/Jng and New Zealand obese (NZO) mice; Kuo Kondo (KK) mice; NoncNZO10/LtJ mice), and genetically-induced models of β-cell dysfunction (human islet amyloid peptide (hIAPP); AKITA and muscle insulin-like growth factor-1 receptor (IGF-1R)-lysine-arginine (MKR) mice) [38,39,40,41,42,44,45].

The C57Bl/6J mouse is the optimal-studied model of diet-induced obesity; however, it could be an acceptable candidate to study DM. The mice do not develop frank T2DM, but rather a mild and transient type. However, it shows high-fat diet/obesity-related musculoskeletal changes, such as lowering of trabecular bone volumes and reduction of bone formation [46]. The MKR exhibits frank hyperglycemia, insulin resistance, and hyperlipidemia by the second week of age. It shows musculoskeletal changes, such as decreased femur stiffness and osteoblastic activity [40]. The Ob/Ob mouse is a model of severe obesity and develops obvious hyperglycemia, -insulinemia, and insulin resistance. It shows a low bone mass in the lumbar vertebrae and the long bones and disturbed physio-biomechanical properties, in addition to shorter femurs in comparison to non-diabetic ones [40]. The db/db mouse is a model of severe T2DM. Its musculoskeletal changes are not clearly addressed; however, disturbances in bone structure and reduction of the length of long bones are reported [47]. Diabetic manifestations are more obvious in males. Nevertheless, the severity of diabetes in both the ob/ob and db/db mice models is dependent on the genetic background: the black Kaliss’s (BKS) mice background results in severe DM and early death. While on the C57BL/6 background, DM is mild and transient [48]. The yellow Kuo Kondo mouse develops severe obesity, insulin resistance, hyperglycemia, and -insulinemia by 8 weeks of age. Its musculoskeletal changes have not been investigated yet [40]. The TallyHo mouse is a model of early-onset, naturally occurring obesity and T2DM. Males of this diabetic model show reduced BMD of the trabecular bone at the distal femur [40]. The M16 mouse is a model of early-onset obesity and shows moderate hyperglycemia [40]. No investigations regarding its musculoskeletal changes were found. The NSY mouse is a model of mild obesity and shows increased glucose levels and hyperinsulinemia [40]. However, no reports of musculoskeletal changes are available. The TSOD develops obesity, impaired glucose tolerance, and hyperinsulinemia [40]. The occurrence of musculoskeletal changes has not been published yet. The NZO mouse exhibits impaired glucose tolerance and could develop frank diabetes. No studies regarding its musculoskeletal changes are described [38,39,40,45].

In contrast to mice, rat models allow more clinical handling and manipulation, so the researcher can repeatedly drain enough blood without affecting the animal, and more tissue samples are available for multiple downstream analysis.

### 1.5. Study’s Basic Idea

We hypothesized that the previously achieved experimental studies in rodent models approved the positive link between T2DM and IVDD, particularly in rodent diabetic models. We undertook a systematic and comprehensive literature review, considering only the published experimental studies, which discussed the link between T2DM and IVDD in rodent diabetic models. Thereby, we also summarized and discussed the different rodent models used. We aimed to achieve the following objectives: (I) achieving a systematic comprehensive literature review of the experimental trials undertaken in diabetic rodent models to study the link between T2DM and IVDD, (II) revealing the strengths and weaknesses of the published studies, (III) combined and comprehensively displaying of the results of the achieved studies in the form of a critical constructive report, and (IV) identifying the unanswered questions in the currently available literature.

## 2. Materials and Methods

### 2.1. Data Search: Literature Sources

We conducted our systematic review and data search and selection according to Preferred Reporting Items for Systematic Reviews and Meta-Analyses (PRISMA) guidelines [49]. We performed an extensive systematic electronic literature search in the available electronic medical databases, including PubMed, PMC, Cochrane Library, Wiley Online Library, Hindawi Publishing Corporation, Hindawi Journals, Web of Science, and Excerpta Medica dataBASE (EMBASE). We used separated keywords regarding the topic, results, methods, etc., such as DM, IVDD, IVD, rats, rat models, spine degeneration, cytokines, NP, AF, GAG, etc. and also titles, such as DM and IVDD, IVDD in rats, animal models in IVDD, etc. After the initial literature research, we performed a records screening, trying to find every relevant record or report from the obtained studies. The list of references for the collected papers was carefully checked to ensure the eligibility of the studies and their accordance with our topic. Each step of the above-mentioned three steps was separately performed by two reviewers (M.M.) and (M.K.), and after that, they were revised and checked by the third and fourth reviewers (A.A.) and (G.S.-T.).

### 2.2. Characteristics of the Initially Obtained Literature

We collected the published studies (9511 papers), which were identified as related to the topic (IVDD). Out of these, 2144 manuscripts were related to the topic IVDD in rodents, 1528 publications related to the topic IVDD in DM, and 27 publications related to our topic IVDD in diabetic rodent models, see Table 1 and Figure 2.

### 2.3. Literature Filtration and Selection

The studies that were included in this review were experimental and had three axes “IVDD, DM, and rodent models”. Therefore, 9485 manuscripts, which were not relevant to our topic, were excluded.

They were either not applicable for this review, or they were not experimental. Finally, after accurate filtration, only 23 publications were included, which were relevant to our topic, see Table 1 and Table 2 and Figure 2**.**

### 2.4. Study’s Admission Requirements

In order to be included, the study should minimally fulfill 5 inclusion criteria, and no exclusion criterion (Table 3). The inclusion criteria were the following: (I) the experimental study should have clearly outlined methods, which investigated effects of (II) T2DM on (III) IVDD in (IV) diabetic rodent models and focused on the pathogenesis, risk factors (RFs), treatment or prophylactic agent of the issue of interest, (V) the publication must be accessible in English with its record and appendices if present. The exclusion criteria were: (I) the study was not experimental (review, clinical, case report, etc.), (II) deficient of DM or (III) IVDD or (IV) rodent model, or (V) its methods were not applicable, unclear, or insufficient.

### 2.5. Data Extraction

The selected studies (23 manuscripts) were also inspected regarding date of publication, publication origin, the focus of study (pathogenesis, risk factor, therapy, and prophylaxis), author details, the aim of the study, methods, type of used rodent model, results, etc. (see Table 4). The included studies were accurately and in detail examined by two reviewers (M.M.) and (M.K.), and the collected data were revised by two leading reviewers (G.S.-T.) and (A.A.).

### 2.6. Evaluation of the Studies

We aimed to evaluate each study regarding its quality and strength by the individual and combined assessment of its components, such as aim, methods, results, etc. In order to achieve this, we adhered to the Newcastle-Ottawa scale (NOS) [50,51]. However, the NOS is highly efficient as an evaluative and analytical tool in the systematic reviews involved in epidemiological and clinical studies, such as cross-sectional studies, cohort studies, etc. Yet, we had to generate our own system, which is a point-based assessment system adapted to experimental studies and applied it in this review (see Section 2.7 and Table 5).

### 2.7. Study Scoring System (SSS)

This system evaluated each study according to 5 components (abstract, aim, research question (RQ), methods, and results). For each one of these components, there were assessment tools/parameters, see Table 5. There were five (5) criteria for abstract evaluation, which comprised clarity, conciseness, readability, completeness, and outline; five (5) criteria for RQ, which were feasibility, interestingness, novelty, ethics, and extent of relevance to our issue; five (5) criteria for aim evaluation were realism, experimental feasibility, clinical applicability, and profitability; five (5) criteria for methods evaluation were appropriateness, efficiency, experimental availability, comprehensiveness, and correct achievement, and five (5) criteria for results evaluation were reliability, absence of bias, validity, applicability, and measurability. The SSS comprises 25 points, 5 points for each item/component. The quality assessment (QA) of each study depends on the gained points in the SSS, as follows: high quality (21–25 points), moderate quality (16–20 points), and low quality (≤15 points).

### 2.8. Evidence Assessment System (EAS)

In addition to the above-described assessment system, we set an additional evaluative parameter to estimate the strength of the detected evidence of DM influence on IVDD (rating in the range from 1 to 8), see Table 5. This system depends on the total sum of the different types of findings. Each finding for an interrelation between DM and IVDD was an indicator and the summation of all indicators, supporting that the hypothesis “DM and IVDD are interrelated in the rodent model” exhibits the final/real evidence of the study. The supporting findings or indicators could be acquired by histopathology (H), immunodetection (I), molecular biology (M), biochemical (BC) or biomechanical methods (B), imaging (IM), clinical (C), and/or statistical (S) approaches; coincidence between the results, underline the relation between IVDD and T2DM in the used rat model-derived tissues and if included, also in human tissues. The evidence strength of IVDD and DM interrelation was documented as following: strong (≥5 indicators), moderate (3–4 indicators), mild (1–2 indicators), and absent (0 indicators). There were nine studies included, which were exclusively undertaken in vitro, and two showed, in addition to in vivo, also in vitro results and described the devastating effect of glucose in culture medium on IVD cells, see Table 4 and Table 5. In these studies, the IVD cells were isolated from non-diabetic rat models, and afterward, they were cultured in growth mediums with elevated glucose content of different concentrations. The rest of the included studies demonstrated the effect of DM on IVD tissues in vivo, as the IVD cells were isolated from diabetic experimental rat models and non-diabetic rat models as a control, see Table 4. Both in vivo and in vitro proceedings, were considered in our study to provide evidence for the link between DM and IVDD with different strengths of the detected evidence of DM influence on IVDD degrees according to the scoring system described above.

## 3. Results

### 3.1. Identified Studies

We identified 9511 published studies related to the issue IVDD. Out of these 27 manuscripts were related to the topic IVDD in diabetic rodent models, which is our research field of interest [3,15,43,47,52,53,54,55,56,57,58,59,60,61,62,63,64,65,66,67,68,69,70], see Table 4 and Table 5. By further evaluation another four publications were excluded; two of them discussed the diabetic animal models used in IVDD, including rodents, but they were not related to IVDD. They focused only on studying the different animals, such as goats, rodents, dogs, etc., which were used as experimental diabetic models [35,36]. The third one was excluded because it discussed the role of adiponectin, mainly found in obese individuals, but it did not address any DM model [71], and the fourth excluded study dealt with a spinal injury in diabetic rats as one unit and not with IVDD as the main subject of this study [72]. After the final careful filtration, 23 studies were selected and included in our review, which were fulfilling the inclusion criteria listed in Table 3**.**

### 3.2. Origin of Included Studies

Regarding the year of publication, 2013 and 2015 had the highest number with four studies, representing 17.4% for each year; after them, 2014 and 2020 represented themselves in the third position with three studies (13.0%) for each country. The publication years (2016, 2017, and 2018) were in the fourth position with two studies (8.7%) per year, while the publication years (2019, 2012, and 1992) were the last in sequence with only one publication (4.34%) for each year, see Table 4 and Appendix A. According to their origin, eight studies were derived from China, representing 34.8% of total included studies, seven studies from Korea, representing 30.4%, five studies from the USA, representing 21.7%, and 3 studies from the USA with Israel, Taiwan, and Japan, representing 4.34% for each country, and no study from Europe, see Table 4 and Appendix A.

### 3.3. Data Assortment, Analysis, and Evaluation

We studied the collected publications to get the information regarding their focus of the trial, type of evidence, quality of the study, and evidence strength, see Table 5 and Appendix A. We found that 14 studies representing 60.9% of the whole included studies in the review performed in vivo experiments in order to test the effect of DM as a metabolic disease on the IVD tissues and whether it is playing a role in developing IVDD using rodent diabetic models. Nine (9) studies, representing 39.1%, were undertaken in vitro and investigated the effect of different glucose concentrations on IVD tissues extracted from non-diabetic rat models. In addition, two studies combined in vivo and in vitro analyses. The controls in the in vivo performed studies were non-diabetic rodent models, while the controls in the in vitro performed experiments were IVD tissues cultured in normal or low glucose concentrations containing growth media, see Table 4**.** The in vivo studies were using mice (5×) or rats (9×), whereas all in vitro studies included were based on rat-derived IVD cells.

Fourteen studies, investigated the overall pathogenesis of DM in the development of IVDD, see Table 4, [53,54,55,56,57,59,61,63,64,65,67,69], while two studies investigated a specific pathogenesis element [58,66], such as the role of Sirt1/p53 axis in diabetic IVDD and the effect of metastasis-associated lung adenocarcinoma transcript 1 (MALAT1) on apoptosis promoted by high glucose in rat cartilage endplate (CEP) cells. The other four studies (17.4%) investigated the effect and pathogenesis of risk factors [3,52,60,70], such as the degenerative effect on spine/IVD tissues induced by chronic ingestion of AGEs and the long exposure to high glucose concentrations. Two studies explored preventive methods/agents (prophylaxis) of IVDD induced by DM, such as vitamin D and anti-AGEs drug [62,68]. One study (8.7%) surveyed the physio-biomechanical properties of the diabetic degenerated IVD [15,54]. One study investigated the pathogenesis of the diabetic IVDD by carrying out a therapeutic trial using IGF-1 injection as an anti-DM drug [47], see Table 4**.**

#### Quality of the Studies Included

According to the assessment of the quality of the included studies using our generated SSS, 12 studies, representing 52.2%, were graded as high in quality, six studies, representing (26.1%), were graded as moderate in quality, and the last five studies (21.7%) were graded as low in quality, see Table 5.

Most data supporting the link between IVDD and T2DM were statistically affirmed and achieved by immunodetection, followed by data based on biochemical and histopathological findings. Moreover, the in vivo studies exhibited one clinical and two biomechanical positive findings. The in vitro studies separately investigated the degenerative changes caused by T2DM in most of the anatomical structures of IVD (AF, NP, CEP, and notochordal cells).

### 3.4. Evidence Assortment and Assessment

Evidence was calculated based on the number of separate experimental approaches within one study, which revealed significant differences between the T2DM group and controls in regard to IVDD development. All studies based on diabetic rodents proofed the diabetic status of the animals by checking the blood glucose levels, mostly measured after fasting (Table 4). Regarding the type of found evidence (Table 5), we identified collectively various related findings/indicators linking DM with IVDD. The related findings included histopathological (H), immunodetection (I), biochemical (BC), biomechanical (B), molecular biological (M), imaging (IM), clinical (C), and statistical (S) proofs. The immunodetection of factors changed due to IVDD combined with T2DM represented the majority of approved experimental results. All studies were included irrespectively of whether performed in vivo or in vitro (Table 4 and Table 5). Nearly all in vivo studies presented histopathology. Since the biomechanical findings were limited to the included in vivo studies, only a few biomechanical investigations were performed [3,43]. If performed, imaging aimed to elucidate bone alterations accompanying IVDD in DM. Clinical features existed in only one included study (withdrawal test to assess pain) [69]. In vitro studies generally could not show histopathology, clinical evidence, and supporting imaging results. Therefore, the in vitro evidence was mostly found on methods based on immunodetection and biochemical assays. Nevertheless, there was a bulk of evidence deduced from the included in vitro studies. Molecular biological techniques were less often applied as a means to achieve evidence for IVDD caused by T2DM [47,59,69].

According to our generated Evidence Assessment System (EAS), see Table 5, evidence was defined in this review as the collective strength of the whole positive findings of each study by different methodological approaches. Four studies were evaluated, possessing strong evidence, representing 17.39% of the total included studies. Seventeen studies were evaluated to possess moderate evidence, representing 73.91% of the whole included studies. One study was evaluated to show low evidence, representing 4.34% of the whole included studies. Nonetheless, no studies showed only one positive finding or negative experimental finding, and only one study (4.34%) reported the absence of evidences [70].

## 4. Discussion

The literature research identified a plethora of published papers related to the topic of IVDD, from which 23 studies connecting to T2DM fulfilled the inclusion criteria. The majority of them (13) was performed in vivo, and the smaller section was based on in vitro testing. A fewer number of studies combined both results from in vivo and in vitro experiments [58,63]. The in vitro studies allowed separate investigation of rodent IVD-derived cell types, such as AF [55], NP [65], notochordal cells [54,59,60], or even EP chondrocytes [66,67], in regard to the link between IVDD and T2DM. Strictly speaking, notochordal cells represent precursors of the NP cells characterized by a specific expression profile and associated with multipotent differentiation and self-renewal capacity and, hence, regenerative potential [5]. The studies related to notochordal cells included here did not further address this specific expression profile, and hence, their discrimination from NP cells remains elusive [54,59,60]. Looking at the in vitro studies when high glucose was used to mimic hyperglycemia conditions, the concentrations applied substantially differ (e.g., 25 mM [65,66]) versus 100, 200, or even 400 mM [54,55,56,59,60] or 5.5, 25, 50, 100, and 150 μM [58]. In this respect, one has to face the fact that the blood glucose levels in diabetic rats (ZDF rats, 4–8 weeks) or mice were between 20 and 30 mM [47,73].

Most of the included studies focused on different aspects of the overall pathogenesis of IVDD in diabetic rodents (e.g., [52,53,54]). Besides the presence of elevated levels of advanced glycation end-products (AGEs) as a result of high glucose concentrations [3,62,63], they hypothesized a deficit in IVD nutrition caused by changes in the thickness and porosity of the bony vertebral EP and, subsequently, reduced microvascularity and vessel diameters in response to T2DM (Figure 3, [53]) as a trigger of IVDD development. Malnutrition also leads to enhanced expression of hypoxia-inducible genes in the IVD [15] (Figure 4).

The high levels of AGEs mediate an intensified interaction with the Receptor for advanced glycation end-products RAGE receptor [69]. AGEs represent pathogenetic risk factors. They are increased in T2DM [15], but they are also risky components of modern diets, and hence, the effect of chronic consumption of diets with high levels of AGE on IVDD was addressed in several of the selected studies [3,52,62,63]. The AGE/RAGE interaction results in amplified autophagy detectable by the expression of diverse autophagy markers in IVD cells [56]. Autophagy is a strategy of stressed cells to eliminate damaged cell organelles. In addition, IVD and NP cells undergo senescence [55,57]. Moreover, mitochondrial stress, characterized by reactive oxygen species (ROS) accumulation but also associated with compensatory elevation of antioxidants, such as catalase and manganese superoxide dismutase (MnSOD), could be detected [54]. As an additional feature of mitochondrial stress, the mitochondrial membrane potential is disrupted [59,67], and a disbalance of mitochondrial B-cell lymphoma 2 (Bcl-2) family apoptosis regulators could be observed [67]. Elevated pro-apoptotic Bax induces the release of cytochrome c from the mitochondrial intermembrane space into the cytosol, and thus, the apoptosis cascade proceeds [74]. Accordingly, increasing the activity of initiator and executioner caspases is observed [67].

IVD cell stress under high glucose conditions is also characterized by the increased release of inflammatory cytokines, such as Tumor Necrosis Factor (TNF)α [61,68], interleukin (IL)-1, and IL-6 [61]. These cytokines contribute to the unrestrained induction of diverse matrix-metalloproteinases (MMPs) and A disintegrin and metalloproteinase with thrombospondin motifs (ADAM-TS), which degrade a wide range of ECM components, leading to the loss of IVD ECM integrity and stability [61,62]. In addition to cell loss by apoptosis, the specific anabolic synthesis profile of IVD cells is impaired by cell stress and senescence in response to high glucose, which further weakens the IVD ECM and, therefore, accelerates its degeneration. With regard to this, the anabolic growth factors—IGF-1 and TGFβ—are reduced, which are known to stimulate ECM synthesis [68]. Thus, IGF-1 treatment is proposed as a strategy to inhibit IVD degeneration under T2DM conditions [47]. The expression of IVD-associated ECM proteins, such as collagen type II, I, GAGs, aggrecan, and the chondrogenic transcription factor SOX9, is suppressed under high glucose [15,62,64]. Moreover, ECM composition might also change due to IVD cell trans-differentiation since hypertrophic markers, such as collagen type X, could be detected in IVD cells in response to hyperglycemia [52]. Another study observed a loss of tissue organization, such as IVD fibrosis, and the absence of the AF lamellar pattern, suggesting that the order of collagen fiber bundles disappeared under hyperglycemia [61]. Therefore, the capability of the IVD to distribute homogeneously the compressive strain might be impaired under these conditions.

Because abundant AGEs associated with T2DM also enhance collagen cross-linking [3,27] responsible for increased IVD stiffness and loss of GAGs and proteoglycans, which impair water binding and viscoelasticity, the overall biomechanical properties and adaptability of IVD are severely altered under T2DM conditions. As a final key feature, this impaired IVD biomechanics ultimately leads to IVDD with increasing the risk of herniation and resulting in radiculopathies [69]. Apart from suggesting the involvement of activated mitogen-activated protein (MAP) kinases (extracellular signal-regulated kinase: ERK and p38) in the interrelation of T2DM and IVDD [58,60,63], signaling pathways shared by T2DM and IVDD pathogeneses were further addressed in detail in the 23 studies included. Only two studies investigated the more specified pathogenesis by addressing the role of Sirt1/p53 interplay in the interrelation between DM and IVDD in regard to NP cell apoptosis [58]. They evaluated the role and the underlying mechanism of MALAT1 in the apoptosis of cartilage endplate (CEP) cells induced by high glucose concentrations [58] providing more detailed analysis of the mode of interaction.

In regard to the pathogenesis of IVDD under T2DM conditions, the resulting hyperglycemia obviously affects the whole motion segment, embracing the vertebrae [52,62,70] and the components of the IVD (AF and NP) between them [43,61]. Despite the present study focused on the IVD, the hyaline cartilaginous part of the EP and vascularization of the bony EP were addressed because both represent an important precondition for IVD nutrition [53].

Two studies presented also putative therapeutical options by treatment with glucose suppressing injection of IGF-1 [47] or with butein as an agonist of protective Sirt1 [58]. Two studies addressed options for IVDD prevention under high glucose conditions. The authors of the first study tested the effect of vitamin D (calcitriol) on IVDD in DM, which could partially restore TGFβ and IGF-1 in the discs [68]. The other research group found that the oral treatment with a combination of anti-inflammatory and anti-AGE drugs (pentosan-polysulfate and pyridoxamine) was able to reduce to some degree diabetes-induced degenerative changes in the vertebra and IVD [62].

Using animal models, sex aspects have to be accounted for. Most of the research groups (Table 4) selected male rodents for their experiments. Unfortunately, some of the summarized studies did not provide any sex information [15,43,52,68]. Sex differences are barely investigated. Mostly male animals were used (18 of 23 studies, four studies provided no sex information). Two research teams compared males and females and found that AGE in diet had more effects in females [3,70], and one study used only female mice [62]. Krishnamoorthy and Natelson et al. compared the effect in males with female mice, elucidating a clear association with more severe effects on IVD biomechanics in females [3] and impaired bone structure [70]. In contrast, regarding a leptin receptor-resistant model, Li et al. stated a protective effect in females due to sex steroids [47] as a justification to include only males in their experimental groups. The observable sex effects might depend on the model (mouse vs. rat, different lines of both species, specific knockout) or also on the mode of T2DM induction used and should necessarily be investigated in more detail in the future.

In regard to the type of rodent in vivo models, rat models dominated (eight of 13 in vivo studies) and were often based on SD rats treated with streptozotocin (STZ) [53,58,63,64,68]. The preference of rats the fact that all in vitro studies were based on rat-derived cells might be due to the larger sample size in rats compared to mice. The preferential use of STZ-treated SD rats compared to genetically-modified rats might be influenced by costs.

Bearing in mind that the spine function depends on its unique biomechanics and T2DM might contribute to biomechanical failure; unfortunately, biomechanical investigations were undertaken by just a few of the selected in vivo studies. Looking at the in vitro studies, all of them used monolayer cultures, but no three-dimensional (3D) culture models were included, which might mimic more closely the in vivo conditions. 3D culture could also allow biomechanical testing in vitro in future DM cell models.

However, this literature review revealed four strong, 17 moderate, and one low evidences of the link between DM and IVDD in rats in vivo and in vitro, which clearly underlines the effect of DM on the development of IVDD, while only one study reported an absence of the evidence [70].

The number of rodents used varied greatly in the in vivo studies, usually ranging between 20 and 48 individuals. One study [75] used a huge number of experimental animals (about 255 rats) divided into two groups. The in vitro studies generally used fewer individuals for cell harvesting. In these studies, performed in vitro, generally, only one group of non-diabetic rats (mostly SD rats) gave the needed IVD cells, which were then incubated in the culture medium of two different glucose concentrations as experimental and control groups. The direct comparison to human samples or IVD cell population and estimation of transferability of results remains mostly unclear (one has to be aware that there are notochordal cells in rodents. Furthermore, in rodent models, often coccygeal IVDs are used because they are larger than lumbar discs in the rodents [15]). Only the in vitro study of Tsai et al. included parallel experiments with human cells extracted from IVD tissues of healthy and diabetic patients, suggesting a nice parallel agreement of the results [63].

Worth to mention is that some crucial approaches, such as biomechanical investigations, are underrepresented in the included pool of studies.

### 4.1. Study Strength

The strength of this study is based on the large number of the included studies, which linked in vivo and in vitro results. In addition, it simultaneously summarized studies, which analyzed the pathogenesis, treatment, prevention, and risk factors of the issue of interest. Altogether, this presents a plethora and wide range of detected positive findings. Moreover, it covers a long period of data generation extending from 1992 to 2020.

### 4.2. Study Limitations

One limitation of our study may be the literature-based study design, whereby the group sizes, the particular type of rodent model, focus, and down-stream analyses of the studies summarized and discussed here differed from each other. It might also hamper the fact that in vivo and in vitro studies were put together to get a sufficient number of relevant studies included.

In addition, mouse and rat models were summarized, which might bear some heterogeneity.

## 5. Conclusions

T2DM and IVDD are likewise diseases with a high prevalence, incidence and distribution, affecting individuals with different socioeconomic levels. Therefore, both have a high public health burden worldwide. Many diseases have exhibited themselves as complications or co-morbidities in diabetic patients. The evidence of the role of DM in the development of IVDD based on rodent models was clearly detected ranging between strong and moderate in power. Hence, based on the collective findings from the experiments performed in vivo and in vitro, our hypothesis regarding the link between T2DM and IVDD is strongly supported. However, collecting a variety of aspects from the available literature that contribute to the linked pathogenesis (Figure 4)**,** the shared signaling pathways behind it are still hidden and need to be explored in the future in order to identify effective therapeutical targets.

## Figures and Tables

**Figure 1 cells-09-02208-f001:**
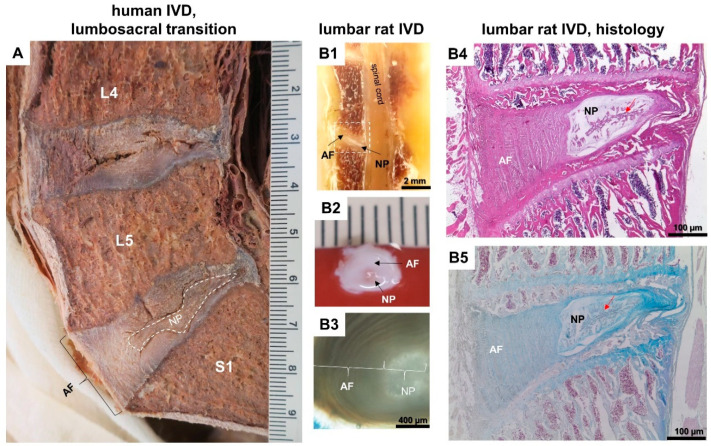
Lumbar intervertebral discs (IVD) of humans and rats. (**A**): degenerated human lumbar IVD—it is Thompson grade 4 or Ray grade 2 since fissures are detectable, extending from NP through the AF [17]. (**B**): Heterozygote Zucker diabetes fatty (ZDF/fa) rat-derived healthy lumbar IVD. (**B1**): Healthy rat IVD in situ (sagittal plane). (**B2**): Explanted rat IVD (transverse plane). (**B3**): Microscopical view of explanted IVD (transverse plane). (**B4,B5**): Longitudinal section (sagittal plane). (**B4**): Hematoxylin-Eosin-stained healthy rat IVD. Red arrow: notochord-derived cells forming band-like structure. (**B5**): Alcian blue-stained rat IVD: glycosaminoglycans are blue. AF, annulus fibrosus, NP, nucleus pulposus. Scale bars (**B1**): 2 mm, (**A**,**B2**): 1 mm, (**B3**): 400 µm, (**B4,B5**): 100 µm.

**Figure 2 cells-09-02208-f002:**
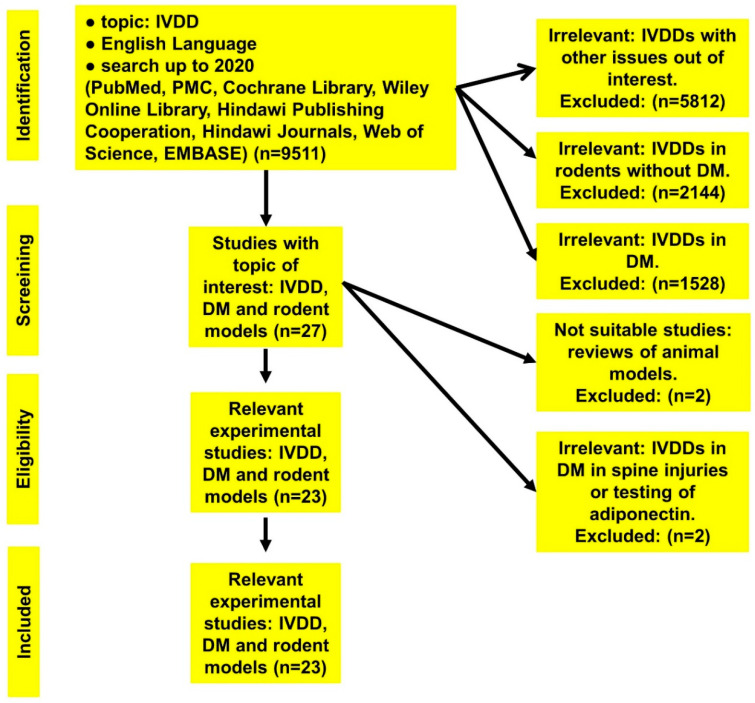
Flowchart of the search strategy and screening process. DM: diabetes mellitus, IVDD: intervertebral disc degeneration.

**Figure 3 cells-09-02208-f003:**
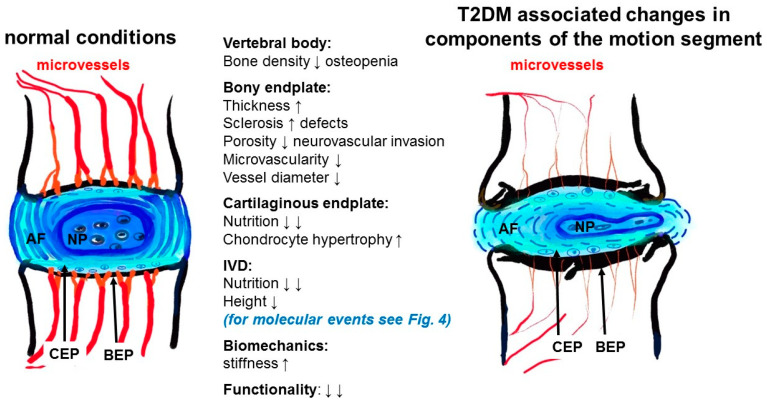
Summary of pathological T2DM-associated changes in components of the motion segment. Based on the literature summarized in Table 4. Molecular events in the IVD are depicted in Figure 4. AF: annulus fibrosus, BEP: bony endplate, CEP: cartilaginous endplate, T2DM: type 2 diabetes mellitus, NP: nucleus pulposus, IVD: intervertebral disc, ↓: decrease, ↑: increase.

**Figure 4 cells-09-02208-f004:**
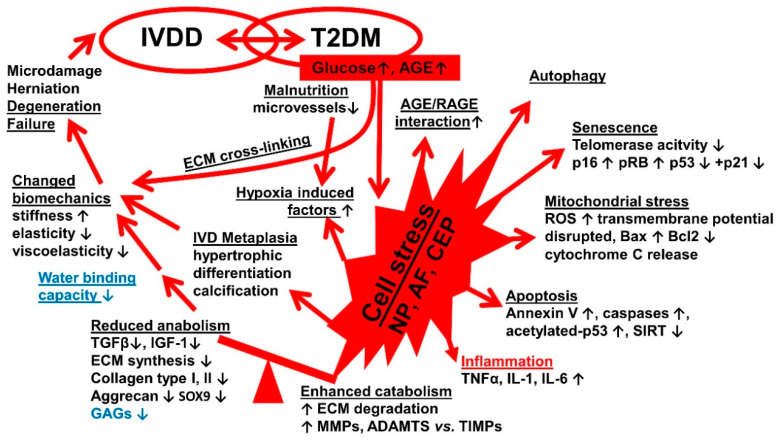
Crucial aspects of the linked pathogenesis of IVDD in T2DM are identified using rodent models. Data depicted here was extracted from studies summarized in Table 4. ADAMTS: a disintegrin and metalloproteinase with thrombospondin motifs, AGE: advanced glycosylation end-products, Bax: Bcl-2 associated protein X, Bcl-2: B-cell lymphoma 2, ECM: extracellular matrix, GAG: glycosaminoglycans, IGF-1: insulin growth factor-1, IL: interleukin, IVDD: intervertebral disc degeneration, MMP: matrix metalloproteinase, RAGE: a receptor for advanced glycation end-products, ROS: reactive oxygen species, SIRT: sirtuin, pRB: retinoblastoma protein, SOX9: sex-determining region SRY of the Y chromosome, TIMPs: tissue-specific inhibitor of matrix metalloproteinases, TNFα: tumor necrosis factor α, T2DM: diabetes mellitus type II, TGFβ: transforming growth factor β.

**Table 1 cells-09-02208-t001:** Classification of the totally collected manuscripts up to May 2020.

Topic	Number of Collected Publications
IVDD	9511
IVDD in rats	2144
IVDD in diabetic cases	1528
IVDD in diabetic rodent models	27

IVDD: intervertebral disc degeneration.

**Table 2 cells-09-02208-t002:** Literature selection.

Studies (n)	Status	Cause	Type
9484	excluded	irrelevant	Reviews, retrospective, and prospective clinical, epidemiological, cross-sectional studies, etc.
2	excluded	not applicable	Reviews of animal models (no experiments)
2	excluded	irrelevant	Experimental:
			1st study: IVDD, rat model, and adiponectin
			2nd study: DM, rat model, and spine injury
23	included	relevant	Experimental (DM, IVDD, and rodents)

DM: diabetes mellitus, IVDD: intervertebral disc degeneration.

**Table 3 cells-09-02208-t003:** Inclusion and exclusion criteria.

Inclusion Criteria (V/V)	Exclusion Criteria (I–V/V)
Experimental with clearly described methods (I)	Non-experimental (review, clinical, case report, etc.) (I)
T2DM (II)	Deficient of DM (II) or IVDD (III) or rodent model (IV)
IVDD (III)	Unclear methods (V)
Rodent models (IV)	
Study of pathogenesis, RF, treatment, prophylactic agent Available in English (V)	
Remark: to be included, the selected papers should fulfill five criteria (V/V)	Remark: one criterion up to four criteria is/are enough to exclude the papers (I-V/V)

DM: diabetes mellitus, IVDD: intervertebral disc degeneration, RF: risk factor.

**Table 4 cells-09-02208-t004:** Included studies (The results and aims are mainly cited in the own words of the original authors).

Authors	Title	Aim	Focus	Rodent Model Type	Methods	Key Results
(1) Ziv et al., 1992 [43] USA & Israel	Physicochemi-cal Properties of the Aging and Diabetic Sand Rat Intervertebral Disc	Understanding of the changes in the physiochemical properties of intervertebral discs (IVDs) in the aged and diabetic rats	Physiochemical properties (in vivo)	180 IVDs were extracted from forty-five desert sand rats (gender information not provided) divided into three equal groups: -Young healthy -Old healthy -Young diabetics	-Rat DM model -Blood glucose and insulin levels -Tissue harvest: lumbar IVD -Metabolic studies	-Discs of young diabetic rats demonstrated decreased hydration, fixed charged density and ability to resist compression under osmotic pressures as compared with the young and healthy discs and were more similar to the discs from old rats and from human -IVD: most affected musculoskeletal tissue in sand rats with aging and DM
(2) Chen et al., 2013 [53] China	The correlation between microvessel pathological changes of the endplate (EP) and degeneration of the intervertebral disc in diabetic rats	Identifying the possible mechanism, by which DM induces degeneration of the INDs with focus on microvessel density (MVD) in the EP	Pathogenesis: microvessel density in the EP (in vivo)	30 three-month-old male adult Sprague Dawley (SD) rats. Rats were divided randomly in two groups (n = 15 rats/group): -Streptozotocin (STZ) -induced DM group -Control group	-Rat DM model -Fasting blood glucose -Tissue harvest (lumbar spine) -Histopathology -Immunohistochemistry: collagen types Ⅰ and Ⅱ, factor Ⅷ-related antigen (von Willebrand Factor, vWF) -Transmission electron microscopy (TEM)	-Expression of collagen type I in the DM group was higher than in controls in contrast to collagen type Ⅱ - vWF was expressed in both, but was low in the DM group - MVD of the DM group was smaller compared to that of the controls -The apoptotic index (AI) in the DM group was significantly higher compared to that of the controls -Negative correlation between the MVD of EP and the AI of notochordal cells -Compared to controls, the EP MVD and the vessel width decreased or disappeared in DM rats
(3) Illien-Junger et al., 2013 [62] USA	Combined Anti-Inflammatory and Anti- Advanced Glycation End-products (AGE) Drug Treatments Have a Protective Effect on IVDs in Mice with Diabetes	Investigation of the effectivity of oral treatments with a combination of anti-inflammatory and anti-AGE drugs in preventing diabetes-induced degenerative changes to the spine (IVD and vertebral bone density)	Prophylaxis (in vivo)	Three age-matched groups of 21 female ROP-Os mice (group size 6-8 animals): -Non-diabetic group -Diabetic group (STZ- induced) -Diabetic mice treated with pentosan-polysulfate & pyridoxamine	-Mice DM model -Blood glucose -Tissue harvest: lumbar spine -Micro-computed tomography (µCT)-Histopathology -Immunohistochemistry: Ne-carboxymethyl lysine [CML], methylglyoxal [MG], tumor necrosis factor (TNF)α, Matrix-metalloproteinase (MMP)-13 and a disintegrin and metalloproteinase with thrombospondin motifs-5 (ADAMTS)-5 -GAG measurement	-Diabetic mice exhibited pathological changes: IVD height↓, vertebral bone mass↓, glycosaminoglycans (GAGs)↓ and morphological alterations of IVDs with focal highly expressed TNFα, MMP-13 and ADAMTS-5 -Accumulation of larger MG amounts suggested that AGE accumulation was associated with these diabetic degenerative changes -Treatment prevented / reduced DM induced degeneration of vertebrae and IVD
(4) Jiang et al., 2013 [64] China	Apoptosis, Senescence, and Autophagy in Rat Nucleus Pulposus Cells: Implications for Diabetic IVD Degeneration	Studying of the mechanisms by which DM aggravates IVDD and discussing of the relationship between autophagy and IDD in NP cells	Pathogenesis (in vivo)	Two groups of thirty-four 2-month-old male SD rats (STZ) -Control (citrate buffer) -diabetic (STZ)	-Rat DM model -Fasting blood glucose (plasma) -Tissue harvest: lumbar discs -Histopathology -TEM -Immunohistochemistry: collagen type II, cleaved caspase-3, p16lnk4A, Microtubule-associated protein 1A/1B-light chain 3 (LC-3) -polymerase chain reaction (PCR): collagen types I, II, aggrecan -Westernblot: caspase-8, -9, -3, p16lnk4A, p62, Beclin-1 -terminal deoxynucleotidyl transferase dUTP nick end labeling (TUNEL) assay	-Higher levels of autophagy in NP cells of diabetic rats than control rats (statistically significant) -Proteoglycan and collagen type II in the ECM and the aggrecan and collagen type II mRNA expression in NP cells of diabetic rats were decreased compared with the control group -DM increased apoptosis of NP cells and led to activations of initiators of intrinsic (caspases-9) and extrinsic (caspase-8) pathways as well as their common executioner (caspase-3) -Cellular senescence was increased about twofold in NP of diabetic rats
(5) Fields et al., 2015 [15] USA	Alterations in IVD composition, matrix homeostasis and biomechanical behavior in the UCD-T2DM rat model of type 2 diabetes	Approving of the role of DM in causation of IVDD and in turn low back pain (LBP)	Pathogenesis Patho-biomechanics (in vivo)	One diabetic and two non-diabetic groups (gender not mentioned): -Six-month-old lean SD rats (“control”), -obese SD rats (“obese”), -UCD-T2DM rats (“diabetic”); n = 6	-Rat DM model -Blood glucose (not fasted) -Tissue harvest: coccygeal -Cell harvest & cell culture -Blood glucose measurement -µCT: EP microarchitecture -Histopathology: EP vascular supply -Biomechanical assessment of creep characteristics -Biochemical analysis: GAGs-Immunoassay: AGE -PCR: genes of ECM homeostasis	-DM: GAG and water contents↓ vertebral EP thickness↑, EP porosity↓, AGE level↑ -Discs from diabetic rats were stiffer and less compressible -Expression of hypoxia-inducible genes↑, catabolic markers↑ in AF and NP from DM rats
(6) Illien-Junger et al., 2015 [52] USA	Chronic Ingestion of Advanced Glycation End Products (AGEs) Induces Degenerative Spinal Changes and Hypertrophy in Aging Pre-Diabetic Mice	Investigation of the role of specific AGE precursors on IVDD and vertebral pathologies in aging mice that were fed isocaloric diets with standard or reduced amounts of MG-derivatives	Risk factor (Modern diets contain high levels of AGEs) (in vivo)	Two groups of aging C57BL/6 mice (gender not mentioned) received: -either a low AGE chow produced without the use of heat (n = 12), or -a low AGE chow supplemented with the synthetic AGE-precursor methylglyoxal-bovine serum albumin (MG-BSA), (n = 9) Mice were insulin resistant but not hyperglycemic	-Mice prediabetic model -Fasting blood glucose -Tissue: lumbar spine -µCT -Histopathology -Immunohistochemistry: MG, CML, TNFα, collagen type X, ADAMTS-5	Chronic exposure to dietary MG/AGEs leads to: -Cortical-thickness and cortical surface area↑ -AGE accumulation & ectopic calcification in vertebral EPs -IVD calcification & hypertrophic differentiation -GAG loss -Accelerated IVDD and vertebral alterations with insulin resistance - IVD height↓ NP cells from dMG+ mice exhibited increased collagen type X staining
(7) Park et al., 2016 [61] Korea	Increased Apoptosis, Expression of Matrix Degrading Enzymes and Inflammatory Cytokines of AF Cells in Genetically Engineered Diabetic Rats: Implication for IVDD	Investigation of the effect of DM on apoptosis, expression of matrix degrading enzymes and inflammatory cytokines in cells of IVDs derived from genetically engineered OLETF (diabetic) and LETO (control) rats	Pathogenesis (in vivo)	Two rat groups: 6-month old male OLETF (diabetic) and LETO (control) rats (10 per each group)	-Rat DM model -Glucose tolerance tested -Tissue harvest: lumbar spine -Histopathology -TUNEL assay (apoptosis: AF cells) -Western blot: MMP-1, -2, -3, -13, tissue inhibitor of metalloproteinase (TIMP)-1, -2 -PCR: IL-1, -6 and TNF-α	-OLETF rats showed increased body weight and abnormal 2-h glucose tolerance tests compared to LETO rats -The AI and the degree of Fas expression by AF and the -Expression of MMP-1, -2, -3, -13, TIMPs-1 and -2 was statistically higher in OLETF rats -Expression of IL-1, -6 and TNF-α was statistically higher in OLETF rats -Histological analysis showed more severe fibrosis and loss of lamellar pattern in AF tissues of OLETF rats
(8) An et al., 2017 [68] China	Vitamin D (calcitriol) improves the content of transforming growth factor (TGF)-β and insulin-like growth factor (IGF)-1 in IVD of diabetic rats	Testing of protective effect of Vit. D against IVDD in DM	Prophylaxis (in vivo)	55 SD rats were divided into three groups (gender not mentioned): -Experimental group (STZ+calcitriol) (n = 20) -Control group (STZ+citrate buffer) (n = 20) -Normal group (citrate buffer) (n = 15)	-Rat DM model -Fasting blood glucose -Tissue harvest: lumbar spine -Histopathology -Immunohistochemistry: TGFβ, IGF-1 -Western blot: TGFβ, IGF-1	-Histology revealed degenerative changes in discs of experimental and control group at three different time points, while there were no changes in discs in normal group -Content of TGFβ & IGF-1 in experimental and control group was significantly lower than in normal group at three different time points, but there were also significant lower values in control compared with the experimental group
(9) Kameda et al., 2017 [69] Japan	Investigation of the effect of diabetes on radiculopathy induced by NP application to the dorsal root ganglion (DRG) in a spontaneously diabetic rat model	Evaluation of the effect of DM on radiculopathy due to lumbar disc herniation (LDH), by investigating pain-related behavior and the expression of TNF-α and growth-associated protein (GAP)43 in type 2 diabetic rats following application of NP to the dorsal root ganglion (DRG)	Pathogenesis (in vivo)	Two groups: -A total of 129 13-weeks old male Wistar rats -A total of 126 13-weeks old, male GK rats. The GK rat is a spontaneous model of T2DM Large-sized test and control group -Surgical NP group -Sham group -Naïve rats	-Rat DM models -Blood glucose: refers to previous study -Tissue harvest: lumbar spine -Measurement of mechanical withdrawal thresholds -Immunohistochemistry: Ionized calcium-binding adapter molecule-1 [Iba-1], TNF-α, RAGE -PCR: GAP43 -Western blot: GAP43	-Mechanical withdrawal threshold significantly declined in the non-DM NP group compared to the non-DM sham group for 28 days, whereas the decline in threshold extended to 35 days in the DM NP group compared to the DM sham group -RAGE and TNF-α expression in DRGs was co-localized in Iba-1 positive cells -Non-DM NP rats had higher TNF-α protein expression levels vs. the non-DM sham rats on day 7, the DM NP group had higher levels vs. the DM sham group on days 7 and 14 -Non-DM NP group had higher GAP43 mRNA expression compared to the non-DM sham group for 28 days, while the DM NP group had a higher level than the DM sham group for 35 days
(10) Krishna-moorthy et al., 2018 [3] USA	Dietary AGE consumption leads to mechanical stiffening of murine IVDs	Testing the hypothesis that chronic consumption of high AGE diets results in sex-specific IVD structural disruption and functional changes	Risk factor (chronic consumption of high AGE diets) (in vivo)	mice model: 21 females and 23 males C57BL/6J mice, each assigned to two isocaloric diet groups, receiving either a low AGE or high AGE chow, generated via high-temperature heating	-Mice DM model (both sexes) -Fasting blood glucose and serum -Tissue harvest: coccygeal and lumbar spine -Histopathology -Biomechanical testing - AGE quantification (western blot: serum, IVD) -Molecular assessment of collagen (collagen peptide hybridization) -Second harmonic generation (SHG) imaging with multi photon laser-scanning microscope: collagen fiber orientation	-High AGE diet resulted in AGE accumulation in IVDs and increased IVD compressive stiffness, particularly in females -IVD biomechanical changes result from increased AGE crosslinking in AF -Increased collagen damage did not appear to influence biomechanical properties -High AGE diet has greater influence on females
(11) Li et al., 2020 [47] China	IVDD in mice with type II diabetes induced by leptin receptor deficiency	Studying of the effects of T2DM on IVDD in leptin receptor-deficient knockout mice model. Observation of the effects of T2DM and glucose-lowering treatment on IVDD by IGF-1 injection	Therapy Specific Pathogenesis (in vivo)	Three groups: -wild-type male C57BL/6J mice, -leptin receptor gene knockout, db/db mice -IGF-1 group Only male mice were included because of the DM protective effect of the female sex steroids	-Mice DM model -Fasting blood glucose -IVD harvest (coccygeal + lumbar) -µCT -Histopathology -Immunohistochemistry: MMP-3 -Western blot: leptin receptor -Tunnel assay: apoptosis -PCR: sex-determining region SRY of the Y chromosome (SOX9), aggrecan, MMP-3	-Blood glucose levels were significantly higher in the db/db mice -T2DM in db/db group showed an association with significantly decreased vertebral bone mass and increased IVDD when compared to WT mice - db/db mice showed a higher percentage of MMP-3 expression and cell apoptosis than wild type mice -IGF-1 treatment partly reversed the findings
(12) Natelson et al., 2020 [70] USA	Leptin signaling and the IVD: Sex dependent effects of leptin receptor deficiency and Western diet on the spine in a T2DM mouse model	Investigating, if obesity and DM type II cause spinal pathology in a sex-specific manner using in vivo diabetic and dietary mouse models	Risk factor (hypercaloric Western diets in cases of T2DM, obesity and leptin receptor deficiency) (in vivo)	Four groups of mice models were used: -Two groups of leptin receptor-deficient mice on a C57BL/6J background (B6.BKS (D)-Leprdb/J (Db/Db) (15 females, 19 males) -Two non-diabetic control groups (21 females, 27 males)	-Mice DM model -Fasting blood glucose -Tissue harvest: lumbar IVD -Histopathology -Metabolic studies, (HbA1c level) -µCT: detection of IVD disc height index (DHI) -Biomechanical examination	-Dietary effects on bone structure in Db/Db mice were sex-dependent and evident in females but not males -IVDs of female (but not male) Db/Db mice exhibited morphological changes, but no IVDD -Leptin receptor deficiency did not cause IVDD in 3 months old mice - DHI was not changed in any group -No biomechanical changes, except diminished torsional properties in leptin deficient mice.
(13) Tsai et al., 2014 [63] Taiwan	AGEs in Degenerative NP with Diabetes	Investigation of the effect of AGEs on the degeneration process in diabetic NP and NP cells in rats and humans	Pathogenesis (in vivo and in vitro) Risk factor Peculiarity: Parallel experiments using human and rat-derived NP cells	Nine 8-week-old male SD-rats were divided into two groups:-Non-diabetic (n = 4) -Diabetic group, STZ (n = 5). Human NP tissue: Diabetic (n = 3) Non-diabetic (n = 3)	-Rat DM model -Fasting blood glucose -Tissue harvest: coccygeal discs -Cell isolation & cell culture -Histopathology -Immunohistochemistry: AGE -PCR: MMP-2, RAGE -Zymography: MMP-2 -Western blot: extracellular signal regulated kinase [ERK]	-Immunohistochemical expression of AGEs was significantly↑ in diabetic human and rat-derived discs - MMP-2↑ and RAGE↑, at both mRNA and protein expression levels and phosphorylated ERK↑ in diabetic NP cells as a response to AGEs (human+rat) -AGEs and DM are obviously associated with IVDD in both humans & rats -Hyperglycemia in diabetes enhances the accumulation of AGEs in the NP and triggers IVDD
(14) Zhang et al., 2019 [58] China	The sirtuin (Sirt)1/p53 Axis in Diabetic IVDD Pathogenesis and Therapeutics	Understanding of the relation between DM and IVDD, in particular the Sirt1/p53 axis in NP cells which may be involved in the pathogenesis of diabetic IDD and may also serve as a therapeutic target for diabetic IDD	Specific pathogenesis Therapy (in vivo and in vitro)	Forty-eight adult male SD rats divided into four groups (12 males for group): -DM (STZ) -DM (STZ)+IDD (AF puncture) -Butein treated DM+IDD-Control Isolated NP cells from young rats exposed to high glucose (5.5, 25, 50, 100, and 150 μM)	-Rat DM model -Blood glucose -IVD harvest (probably coccygeal) -NP cell isolation & cell culture -Cell viability assay (cell counting kit-8) -MRI -Histopathology -Immunohistochemistry: Sirt1, acetyl-p53, cleaved caspase-3 -Western blot: Bax, Bcl-2, acetyl p53/p53, p16INK4a, p21WAF1, Sirt1 -TUNEL Assay -Sirt1 Expression & Activity Assay -Reactive Oxygen Species Assay	-High glucose may promote the incidence of apoptosis and senescence in NP cells in vitro -Acetylation of p53 was found increased in diabetic NP cells in vitro. -Hyperglycemia could suppress the expression and activity of Sirt1 in NP cells (in vitro and in vivo) -Butein may inhibit acetylation of p53 and protect NP cells against hyperglycemia-induced apoptosis and senescence through Sirt1 activation
(15) Park et al., 2013 [54] Korea	High glucose-induced oxidative stress promotes autophagy through mitochondrial damage in rat notochordal cells	Evaluation of the effects of high glucose concentrations (0.1, 0.2 M glucose) on the induction of oxidative stress and autophagy through mitochondrial damage in rat notochordal cells	Pathogenesis (in vitro)	Only one non-diabetic group (four-week-old male SD rats), from which the NPs have been harvested before exposed to hyperglycemic or normoglycemic conditions.	-Rat DM model -Tissue harvest: lumbar spine -NP cell isolation and culture under hyper- or normoglycemic conditions -Western blot: autophagy markers -Immunofluorescence: detection of mitochondrial damage and of manganese superoxide dismutase (MnSOD) -Flow cytometry: Intracellular reactive oxygen species (ROS) -Measurement of catalase level	-An enhanced disruption of mitochondrial transmembrane potential, which indicates mitochondrial damage, was identified in rat notochordal cells treated with both high glucose concentrations. -Both high glucose concentrations increased production of ROS by rat notochordal cells in a dose- and time-dependent manner -Two high glucose solutions also enhanced in rat notochordal cells in a dose- and time-dependent manner: -Compensatory expressions of anti-oxidative MnSOD and catalase -Ratio of autophagy markers (LC3-II/LC3-I)
(16) Park et al., 2013 [60] Korea	Dose- and time-dependent effect of high glucose concentration on viability of notochordal cells and expression of matrix degrading and fibrotic enzymes	Understanding of the effect of the duration and severity of DM (using high glucose concentrations: 0.1, 0.2, 0.4 M glucose) on viability of notochordal cells and IVDD	Risk factor (Duration and severity of DM) (in vitro)	Only one non-diabetic group (four-week-old male SD rats), from which the NPs have been harvested before exposed to hyper- or normoglycemic conditions.	-Rat DM model -Lumbar IVD harvest -Cell isolation and culture under hyper- or normoglycemic conditions -Immunohistochemistry -Evaluation of cell proliferation -Evaluation of cell apoptosis (DNA stain) -Western blot: MMP-1,2,-3,-7,-9,-13, TIMP-1,-2, caspase-3, -9, cytochrome-c, Akt, Phospho-p38 MAPK and poly (ADP) ribose polymerase (PARP)	-High glucose significantly decreased proliferation and increased apoptosis of notochordal cells from culture days one to seven in a dose-dependent manner -Compared with normoglycemic group, caspase-9 & -3 activities and cleavage of Bid and cytochrome-c were significantly increased in each three high glucose concentrations, accompanied by increased expression of MMP-1, -2, -3, -7, -9 and -13 and TIMP-1 & -2
(17) Kong et al., 2014 [56] Korea	High Glucose Accelerates Autophagy in Adult Rat IVD Cells	Investigation of the effect of high glucose (0.1, 0.2 M) on autophagy in adult rat AF and NP cells	Pathogenesis (in vitro)	One group of 24-week-old male SD rats, from which the NP and AF cells have been harvested before exposed to hyper- or normoglycemic conditions.	-Rat DM model -Tissue harvest: lumbar spine -Cell isolation and culture under hyper- or normoglycemic conditions -Western blot: autophagy markers (beclin-1, LC3-I and LC3-II, and Atg 3, 5, 7, and 12)	-High glucose significantly increased the expressions of autophagy markers beclin-1, LC3-II, Atg3, 5, 7, and 12 in adult rat NP and AF cells in a dose- and time-dependent manner -Ratio of LC3-II/LC3-I expression↑ (dose- respectively time-dependently)
(18) Park et al., 2014 [55] Korea	Accelerated premature stress-induced senescence of young AF cells of rats by high glucose-induced oxidative stress	Investigation of the effect of high glucose (0.1, 0.2 M glucose) on mitochondrial damage, oxidative stress and senescence of young AF cells	Pathogenesis (in vitro)	Only one non-diabetic group (four-week-old male SD rats), from which the AFs have been harvested before exposed to hyper- or normoglycemic conditions.	-Rat DM model -Lumbar IVD harvest -Cell isolation and culture under hyper- or normoglycemic onditions -Immunohistochemistry: p16, retinoblastom (pRB), p53, p21 -Mitochondrial damage (with Mitotracker in Mitochondrion-selective Probes) -Intracellular ROS measurement with H2DCF-DA -SA-β-Gal activity assay -Telomerase activity using a TeloTAGG (PCR/ELISA)	-High glucose enhanced in a dose- and time-dependent manner: -Mitochondrial damage in young rat AF cells, resulting in enhanced ROS release for one and three days compared to normal control -Senescence of young AF cells -Telomerase activity declined in a dose- and time dependent manner Compared to controls hyperglycemia -Increased the expressions of p16 and pRB proteins and decreased that of p53+p21 in young rat AF cells for one and three days
(19) Kong et al., 2014 [57] Korea	Effect of High Glucose on Stress-Induced Senescence of NP Cells of Adult Rats	Investigation of the effect of diabetes mellitus (DM) on senescence of adult NP cells	Pathogenesis (in vitro)	One group of 24-week-old male SD rats, from which the NP cells have been harvested before exposed to hyper- or normoglycemic conditions.	-Rat DM model -Tissue harvest: lumbar spine -Cell isolation and culture under hyper- or normoglycemic conditions -Immunohistochemistry: p53, p21, pRB, and p16 -SA-β-Gal activity assay	High glucose: -increased the mean SA-β-Gal-positive cell percentage in adult rat NP cells dose- and time-dependently -Increased p16 and pRB and -impaired p53 and p21 proteins in adult rat NP cells
(20) Park et al., 2015 [59] Korea	Rat Notochordal Cells Undergo Premature Stress-Induced Senescence by High Glucose	Investigation of the effect of high glucose (0.1, 0.2 M) on premature stress-induced senescence of rat notochordal cells	Pathogenesis (in vitro)	One group of 4-week-old male SD rats, from which IVD notochordal cells have been harvested	-Rat DM model -Tissue harvest: lumbar spine -Notochordal cell isolation and culture under hyper- or normoglycemic conditions -Mitochondrial damage of notochordal cells (mitochondrial transmembrane potential and apoptosis detection kit) -Intracellular ROS measurement with H2DCF-DA -PCR/ELISA: Telomerase activity -Immunohistochemistry: MnSOD, p53, p21, pRB, and p16 -Expression of catalase -SA-β-Gal activity (SA-β-Gal staining kit)	High glucose enhanced in notochordal cells at 1 and 3 days: -Disruption of mitochondrial transmembrane potential and excessive generation of ROS. -Expressions of MnSOD and catalase - Occurrence of stress-induced senescence by p16-pRB pathways -Telomerase activity declined under high glucose conditions at 1 and 3 days
(21) Cheng et al., 2016 [65] China	High Glucose-Induced Oxidative Stress Mediates Apoptosis and ECM Metabolic Imbalances Possibly via p38 MAPK Activation in Rat NP Cells	To investigate whether high glucose-induced oxidative stress is implicated in apoptosis of rat NP cells and abnormal expression of critical genes involved in the metabolic balance of ECM	Pathogenesis (in vitro)	One 12-week-old male Wistar rats model group, from which NPs were harvested High glucose (5, 15, 25 mM)	-Rat DM model -Tissue harvest: lumbar discs -Cell isolation & culture -Flow cytometry: Annexin V+propidium iodide (PI) (apoptosis) -Measurement of intracellular ROS -Determination of NP cells viability -Western blot analysis (p38 kinase activation) -PCR: collagen type II, aggrecan, SOX9, MMP-3, TIMP-1	High glucose -Reduced viability of NP cells, induced apoptosis -Resulted in increased ROS generation and p38MAPK activation. -Negatively regulated the expression of type II collagen, aggrecan, SOX9, and TIMP-1 and positively regulated MMP-3 gene expression
(22) Jiang et al., 2018 [67] China	High Glucose-Induced Excessive ROS Promote Apoptosis Through Mitochondrial Damage in Rat CEP cells	Evaluation of the effects of high glucose (0.1, 0.2 M) on CEP cells and to identify the mechanisms of those effects	Pathogenesis (in vitro)	A group of three 6-month-old male SD rats, from which CEPs were harvested -Negative control -0.1 M glucose -0.2 M glucose -0.2 M glucose + alpha-lipoic acid (ALA) 0.15M	-Rat DM model -Tissue harvest: lumbar discs -Cell isolation & culture -Flow cytometry: Intracellular ROS, apoptosis -Mitochondrial Membrane Potential (fluorescence microscopy) -Western blot: cleaved caspase-3, cleaved caspase-9, Bcl-2, Bax, and cytochrome c	-High glucose significantly increased apoptosis and ROS accumulation in CEP cells in a dose- and time-dependent manner. High glucose: -Induced a disrupted mitochondrial membrane potential. -Cleaved caspase-3, cleaved caspase-9, Bax, and cytochrome c↑ but anti-apoptotic protein Bcl-2↓ -ALA inhibited the expression of cleaved caspase-3, cleaved caspase-9, Bax, and cytochrome c but enhanced the expression of Bcl-2 and prevented the membrane potential disruption and apoptosis
(23) Jiang et al., 2020 [66] China	Long non-coding RNA metastasis associated lung adenocarcinoma transcript 1 (MALAT1) promotes high glucose-induced rat cartilage EP cell apoptosis via the p38/MAPK signaling pathway	Evaluation of the roles of MALAT1 in the apoptosis of CEP cells induced by high glucose (25 mM) and to explore the mechanisms underlying this effect	Specified Pathogenesis (in vitro)	A group of three 12-week-old male SD rats, from which CEPs were harvested -High glucose, -High glucose + MALAT1 -Negative control -High glucose + MALAT1 RNAi, -Normal control	-Rat DM model -Tissue harvest: lumbar discs -Cell isolation & culture -RNA interference/cell transfection -Flow cytometry: apoptosis, AnnexinV, PI -Western blot: p38 activation-PCR: MALAT1	-Results revealed that high glucose concentration promoted apoptosis and enhanced expression of MALAT1 in CEP cells. -MALAT1 knockout decreased the expression levels of total and phosphorylated p38 and reduced the apoptosis of rat CEP cells. -Results obtained in the present study indicated that MALAT1 may serve as an important therapeutic target for curing or delaying IVDD in patients with DM

Color code: white: in vivo studies, dark grey: combined in vitro/in vivo studies, grey: in vitro studies. Remark: since all studies comprised statistical analysis, it was not listed under “methods”. ADAMTS: A disintegrin and metalloproteinase with thrombospondin motifs, AGE: Advanced glycation end-products, AI: apoptotic index, Atg3: autophagy related 3, Bax: Bcl-2 associated protein X, Bcl-2: B-cell lymphoma 2, BSA: bovine serum albumin, CEP: cartilage endplate, CML: Ne-carboxymethyl lysine, DM: diabetes mellitus, DHI: Disc height index, DRG: dorsal root ganglion, ECM: extracellular matrix, EP: endplate, GAG: glycosaminoglycans, GAP43: Growth-associated protein 43, GK rat: Goto-Kakizaki rat, Iba1: Ionized calcium-binding adapter molecule-1, IGF-1: insulin-like growth factor, IL: interleukin, IVD(D): intervertebral disc (degeneration), MALAT1: Long non-coding RNA metastasis associated lung adenocarcinoma transcript 1, MG: Methylglyoxal, MMP: matrix-metalloproteinase, µCT: micro-computer tomography, MnSOD: Manganese superoxide dismutase, MVD: microvessel density, PI: propidium iodide, pRB: retinoblastoma protein, RAGE: Receptor for advanced glycation end-products, ROS: reactive oxygen species, ROP-O: radiation-induced oligosyndactyly mice, SD: Sprague Dawley rats, STZ: Streptozoicin, SOX9: Sex-determining region SRY of the Y chromosome, TEM: Transmission electron microscopy, TIMP: Tissue inhibitor of metalloproteinases, TNF: tumor necrosis factor, TGF: transforming growth factor, TUNEL, vWF: von Willebrand factor, WT: wild type.

**Table 5 cells-09-02208-t005:** Study evaluation with SSS.

Authors	Abstract	Research	Aim	Methods	Results	Scoring	Quality	Evidence Strength of	
		Question						IVDD and DM Relation	
(1) Ziv et al., 1992 [43]	3	5	5	3 (in vivo)	3	19	Moderate	3 (BC,B,S)	Moderate
(2) Chen et al., 2013 [53]	4	5	5	5 (in vivo)	5	24	High	3 (H,I,S)	Moderate
(3) Illien-Junger et al., 2013 [62]	4	5	5	4 (in vivo)	4	22	High	5 (H,I,BC,IM,S)	Moderate
(4) Jiang et al., 2013 [64]	3	5	5	5 (in vivo)	5	23	High	5 (H,I,BC,M,S)	Moderate
(5) Fields et al., 2015 [15]	3	5	4	5 (in vivo)	5	22	High	7(H,I,BC,M,B,IM,S)	Strong
(6) Illien-Junger et al., 2015 [52]	3	5	5	5 (in vivo)	5	23	High	5 (H,I,BC,IM,S)	Moderate
								Analyses partly only qualitative	
(7) Park et al., 2016 [61]	5	4	4	3 (in vivo)	4	20	Moderate	5 (H,I,BC,M,S)	Strong
(8) An et al., 2017 [68]	4	5	5	4 (in vivo)	4	22	High	3 (H,I,S)	Moderate
(9) Kameda et al., 2017 [69]	5	5	5	4 (in vivo)	4	23	High	3 (I,C,S)	Moderate
(10) Krishnamoorthy et al., 2018 [3]	3	4	4	3 (in vivo)	4	18	Moderate	4 (H,I,B,S)	Moderate
(11) Li et al., 2020 [47]	5	5	5	5 (in vivo)	5	25	High	6(H,I,BC,M,IM,S)	Strong
(12) Natelson et al., 2020 [70]	5	3	3	3 (in vivo)	4	18	Moderate	0	Absent
(13) Tsai et al., 2014 [63]	4	5	5	5 (in vivo and in vitro)	4	23	High	4 (I,BC,M,S)	Moderate
(14) Zhang et al., 2019 [58]	3	4	4	4 (in vivo and in vitro)	4	19	Moderate	5 (H,I,BC,IM,S)	Strong
(15) Park et al., 2013 [54]	4	4	4	5 (in vitro)	4	21	High	3 (I,BC,S)	Moderate
(16) Park et al., 2013 [60]	5	2	2	3 (in vitro)	3	15	Low	3 (I,BC,S)	Moderate
(17) Kong et al., 2014 [56]	5	2	2	2 (in vitro)	3	14	Low	2 (I,S)	Low
(18) Park et al., 2014 [55]	5	2	2	3 (in vitro)	3	15	Low	3 (I,BC,S)	Moderate
(19) Kong et al., 2015 [57]	5	2	2	2 (in vitro)	3	14	Low	3 (I,BC,S)	Moderate
(20) Park et al., 2015 [59]	5	2	2	4 (in vitro)	3	16	Moderate	3 (I,BC,S)	Moderate
(21) Cheng et al., 2016 [65]	4	4	4	5 (in vitro)	4	21	High	4 (I,M,BC,S)	Moderate
(22) Jiang et al., 2020 [67]	3	4	4	4 (in vitro)	4	15	Low	3 (I,BC,S)	Moderate
(23) Jiang et al., 2018 [66]	4	4	4	5 (in vitro)	5	22	High	3 (I,M,S)	Moderate
Remarks:									
1. Criteria for abstract evaluation: clarity, conciseness, readability, completeness, and outline.									
2. Criteria for the research question (RQ): feasible, interesting, novel, ethical, and relevant.									
3. Criteria for aim evaluation: realistic, experimentally possible, clinically applicable, and profitable.									
4. Criteria for methods evaluation: appropriate, efficient, experimentally available, and comprehensive.									
5. Criteria for results evaluation: reliability, absence of bias, validity, applicability, and measurability.									
6. Study scoring system (SSS): it is a point-based system, which includes 25 points, 5 points for each item.									
7. Quality assessment (QA): depends on the gained points in the SSS: high quality (21–25 points); moderate quality (16–20 points); low quality (≤ 15 points).									
8. Evidence of IVDD and DM relation: strong (≥ 5 indicators), moderate (3–4 indicators), low (1–2 indicators), and absent (0 indicators). Indicators are histopathology (H), immunohistochemistry (I), molecular biology (M), biochemistry (BC), imaging (IM), biomechanics (B), statistics (S), the coincidence between results in rat model tissues and human tissues, reversal or inhibition of pathological findings in IVDD by tested therapies and/or clinical manifestation of IVDD in rats in the incubation period before scarification (C).									
9. The evidences of the studies, which were achieved in vitro, are related mainly to the effect of glucose on IVD cells in vitro, not to the effect of DM as a disease on IVD tissues in vivo/rat models.									

H: histopathology (tissue alterations detected by histological staining), I: immunodetection (antibody-assisted detection of tissue/cell components, including immunohistochemistry, flow cytometry, Western blot, ELISA), M: molecular biology (PCR), BC: biochemistry (e.g., TUNEL, GAG, and other assays, such as proliferation, IM: imaging (µ-CT, MRI, X-ray), B: biomechanics (biomechanical testing procedures), S: statistics (statistical tests), C: clinical manifestation (pain tests, etc.).

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
