# Peer review of "The Relationship between Diabetes Mellitus Type II and Intervertebral Disc Degeneration in Diabetic Rodent Models: A Systematic and Comprehensive Review"

_cells, 2020, doi:10.3390/cells9102208_

Round 1
Reviewer 1 Report
The review by Mahmoud et al. addresses an important topic in intervertebral disc research since review about the intervertebral disc specific relationship with diabetes mellitus type 2 has not been performed yet. However, the authors do not evaluate the quality of the findings from the chosen studies, but give a detailed description about how many studies where “classified as high” or “low”. Describing the actual findings of the high/low quality publications in the text (rather than in tables) would improve the manuscript and could make a significant contribution to the field. The result section could be shortened by moving most of the text into tables/supplemental tables.
General comments:
Section 1.1
- Including the anatomy of the spine might not be necessary in a review about the relationship pf diabetes and IVDs. I suggest to start with the motion segments.
- Please re-write the section about the IVD composition. Some statements about IVD composition are not correct as written (please see Urban+ Amer.zool., 40:53-61 2000). The human IVD is only hydrogel like in Thompson grade 1 and becomes more fibrous afterwards. Only a healthy disc NP is white.
- What is meant with “it forms no border”? Do you mean the transition zone between NP and AF? Figure 1 does not represent what is described in the text. (no whitish, hydrogel like IVD), both IVDs (Figure 1A) are degenerated with fissures extending from NP through the AF (Thompson grade 4).
Section 1.2
Please include all types of diabetes (i.e. gestational, secondary and genetics).
Section 1.3
- Line 104 – 115: Those studies do not seem to be correlated with diabetes. Please remove them from this section. Maybe integrate them in section 1.1.
- Line 152-160: Please include the info about the different mice models in the mice model section.
Section 1.4:
- Line 200 and 203: The severity of diabetes in ob/ob and db/db mice is depending on the genetic background: the BKS background results in severe diabetes and early death. While on the C57BL/6 background, diabetes is mild and transient (e.g. https://www.jax.org/strain/000632). Please correct this section.
- Line 220-224: Please include this in the rat section and remove from here.
Section 2.3
- What do you mean with “full text weren’t accessible”? Where those papers not open access (i.e. you would have to buy them)? If that was the reason, then you have to include them. The paragraph needs to be re-written i.e. “shouldn’t” has to be replaced with e.g. “had to be” or “must”.
Section 3.1-3.3.1
- The result section does not provide any information about the details from the different studies. Please convert the information of the tables into the main text and reformat the current text into tables
Section 4:
- Please structure the discussion by the importance of the paper
Specific comments:
- Line 43: It should be IVDs lay between the vertebrae
- Line 89: There is still some controversy about the effect of diabetes on IVD, please soften your language (Alpantaki+ Eur Spine J 2019).
- Line 104: Please provide a reference for the Raj classification.
- Line 121: Please use a more scientific language (not “isn’t so easy”).
- line 124: What do you mean with “invested costs to get statistically significant results”? Do you mean the costs for the number of animals that have to be used to get meaningful results? Please rephrase.
- Line 125: There are more reviews that compare animal models (e.g. Alini+ Eur Spine J 2007).
- Line 127: Please include references for the different models.
- Line 135: Please change fetal life to prenatal
- Line 137: Please remove “directly”
- Line 141: Please change “couldn’t” to “could not” contractions should not be used in written English (please change this through the manuscript).
- Lines 144-150: Please provide reverences for all methods.
- Line 152: include references for “numerous methods”
- Line 182: Please spell out Streptozotocin and mention that it is usually used for T1DM.
- Line 195: “isn’t a bad model” is not scientific. Please use different word choice. Do you mean that it is a “bad” model? What do you mean with “clear diabetes”? Please be more specific.
- Line 253+255: please remove “huge”, “various” and “finally”, these words are not scientific in this content.
- Line 454: This is the first time AGEs
Figures/Figure captions
- Figure 1: A) the bracket for the AF only points to the inner AF, not the outer AF, please mention the degeneration grade and add an image of a non-degenerated human IVD. B1) please use a higher magnification image. B4) The red arrow points to the cartilaginous EP, not the notochordal like cells. Please mention which plane is shown (sagittal, transverse). Orient the images that the caudal site is on the bottom (at the moment the images B4+5 are upside down). Please add scale bars.
- Table 3: what does NB mean?
- Table 5: include the first authors (not only the citation) in column 1
Author Response
Reviewer 1
Comments and Suggestions for Authors
The review by Mahmoud et al. addresses an important topic in intervertebral disc research since review about the intervertebral disc specific relationship with diabetes mellitus type 2 has not been performed yet.
However, the authors do not evaluate the quality of the findings from the chosen studies, but give a detailed description about how many studies where “classified as high” or “low”.
Response: We should differentiate between two different issues: 1) the quality of the study and 2) the strength of evidence, see table 5. The given quality of the included studies was based on a scoring system, which depends on the evaluation of the whole parts (abstract, aim, research question, methods, and results) of each study and the total score of the previously mentioned items gives each study its quality level, while the strength of evidence of each study depends on the total score of the collected positive evidences/findings. The included proofs must be one of the following determined finding/evidence types (statistical, clinical, immunological, imaging, biomechanical, biochemistry, molecular biology or histopathology). Only, the proofs of accepted quality according to the assessment criteria were included in the assessment, while the poor evidences were discarded.
Describing the actual findings of the high/low quality publications in the text (rather than in tables) would improve the manuscript and could make a significant contribution to the field.
Response: We considered this advice, but we found that the description of the findings in text - even in brief - of all included studies would extremely enlarge the manuscript without further gain in information, particularly, the findings were well explained in the table 4. The results summarized in table 4 are discussed in detail in the discussion section.
The result section could be shortened by moving most of the text into tables/supplemental tables.
Response: The result section was tailored according to the description system of reviews to go with the obtained data point for point to cover and outline all of the needed parts and to give the reader the complete idea/picture for better understanding. It was also structured to be the wide and paved entrance to the discussion and conclusion. Also, the described parts under the result section are mostly included in tables, e.g. the part 3.1. topic “identified studies” is summarized in figure 2 and included in tables 3, 4 & 5; the part 3.2. topic: origin of included studies is mentioned in tables 4 and in supplemental figure 1 (diagram 1A & 1B); the part 3.3. topics: data assortment, analysis & evaluation are included in tables 4 & 5 and in supplemental figure 1; the part 3.4. topics “evidence assortment and assessment” are included in tables 4 & 5.
General comments:
Section 1.1
- Including the anatomy of the spine might not be necessary in a review about the relationship of diabetes and IVDs. I suggest to start with the motion segments.
Response: We agree with the reviewer and removed the first paragraph describing spine anatomy of the introduction/basics from line 4 to line 12.
- Please re-write the section about the IVD composition. Some statements about IVD composition are not correct as written (please see Urban+ Amer.zool., 40:53-61 2000). The human IVD is only hydrogel like in Thompson grade 1 and becomes more fibrous afterwards. Only a healthy disc NP is white.
Response: We state now that it is only true in the healthy intervertebral disc (lines 16-17).
- What is meant with “it forms no border”? Do you mean the transition zone between NP and AF? Figure 1 does not represent what is described in the text. (no whitish, hydrogel like IVD), both IVDs (Figure 1A) are degenerated with fissures extending from NP through the AF (Thompson grade 4).
Response: “it forms no border” means there isn’t a demarcation zone. Hence, we write now: “both parts are connected by the transition zone“ (lines 16-17).
We explain now in the Legend of Figure 1 that the depicted human IVD is degenerated and can be classified into Thompson grade IV.
Section 1.2
Please include all types of diabetes (i.e. gestational, secondary and genetics).
Response: Initially, we only focused und mentioned the most common two types, which are related to our topic and could be induced in rodents in lab for scientific researches. Now, we added a more global explanation of DM in 1.2 (lines 36-44):
“DM is classified into four major types T1DM, T2DM, gestational diabetes and specific DM. T1DM and T2Dm are the most common types and will be discussed below, while specific/secondary and gestational diabetes are less common. Gestational diabetes mellitus (GDM) is defined as any glucose intolerance by the onset or during the course of pregnancy, even /regardless if it resolved or remained after delivery. Specific types of DM could be genetic, because of genetic defects beta-cell function or insulin action and could be secondary to diseases of exocrine pancreas, endocrinopathies, drugs and/or chemicals, infections, autoimmune disorders or secondary to genetic syndyromes associated with DM.”
Supporting this classification we cited the references:
- Expert Committee on the Diagnosis and Classification of Diabetes Mellitus. Diagnosis and Classification of Diabetes Mellitus. American Diabetes Association. January 2014, Diabetic Care, care.diabetesjournals.org, Volume 37, Supplement 1, pages: S81-S90. Available online at DOI: 10.2337/dc14-S081. © 2014 by the American Diabetes Association. See http://creativecommons.org/licenses/bync-nd/3.0/ for details.
- Expert Committee on the Diagnosis and Classification of Diabetes Mellitus. Diagnosis and Classification of Diabetes Mellitus. American Diabetes Association, Diabetes Care, care.diabetesjournals.org, Volume 33, Supplement 1, January 2010, pages: S62-S69. Available online at DOI: 10.2337/dc10-S062 © 2010 by the American Diabetes Association. Readers may use this article as long as the work is properly cited, the use is educational and not for profit, and the work is not altered. See http://creativecommons. org/licenses/by-nc-nd/3.0/ for details.
Section 1.3
- Line 104 – 115: Those studies do not seem to be correlated with diabetes. Please remove them from this section. Maybe integrate them in section 1.1.
Response: Since the above mentioned line reference does not match anymore I assume that line 68 is meant. We transferred the introducing sentences of 1.3 to the beginning of section 1.1 (lines 4-8). The grading systems delineated at the end of 1.3. has been better connected to DMT2 by a novel bridging sentence (lines 82-84). We introduced also the Thompson grading system as used by the reviewer and supported it with references.
- Line 152-160: Please include the info about the different mice models in the mice model section.
Response: The reviewer may mean lines 130-144: This paragraph has been restructured, supplemented with references and seems to be harmonious now with the rest of the text. It gives only a short introduction into the models and hence, we decided not to relocate it.
Section 1.4:
- Line 200 and 203: The severity of diabetes in ob/ob and db/db mice is depending on the genetic background: the BKS background results in severe diabetes and early death. While on the C57BL/6 background, diabetes is mild and transient (e.g. https://www.jax.org/strain/000632). Please correct this section.
Response: We changed the mentioned text regarding ob/ob and db/db and regarding C57/BL/6 accordingly (lines 179-181 and 191-193). We thank the reviewer for his expert comment and inserted it in this paragraph for more clearness.
- Line 220-224: Please include this in the rat section and remove from here.
Response: The reviewer may mean line 210-212. We removed this sentence and hence, mice and rat models are clearly separated in this paragraph now.
Section 2.3
- What do you mean with “full texts weren’t accessible”? Where those papers not open access (i.e. you would have to buy them)? If that was the reason, then you have to include them. The paragraph needs to be re-written i.e. “shouldn’t” has to be replaced with e.g. “had to be” or “must”.
Response: We thank the reviewer for indicating this mistake. The phrase “full text was not accessible” has been removed, because it was not correct. The word “should” has been replaced by “must”. Accordingly, table 3 (inclusion and exclusion data) was adapted.
Section 3.1-3.3.1
- The result section does not provide any information about the details from the different studies. Please convert the information of the tables into the main text and reformat the current text into tables
Response: Further descriptions of the studies mentioned in table 4 would extremely enlarge the manuscript without further gain in information. The discussion section explains the results summarized shortly in the tables of the result section.
Section 4:
- Please structure the discussion by the importance of the paper
Response: The discussion section is structured according to the topics of the studies (e.g. pathogenesis, prophylaxis e.t.c.. Unfortnately, high evidence and high quality levels of the included studies go not along with each other. Hence, we decided not to change the order of the discussion according to importance of manuscripts to maintain the thematic coherence.
Specific comments:
- Line 43: It should be IVDs lay between the vertebrae
Response: The reviewer may intend/mean line 13: we corrected it accordingly.
- Line 89: There is still some controversy about the effect of diabetes on IVD, please soften your language (Alpantaki+ Eur Spine J 2019).
Response: The reviewer may intend/mean the phrase now transferred to line 6: The word “strikes” has been replaced by the word “occurs”, however, this paragraph speaks about LBP in general and not in relation to DM. The discussion about the link between LBP/IVDD and DM begins with the line 68 and has been more softly expressed now.
- Line 104: Please provide a reference for the Raj classification.
Response: The reviewer may mean line 82-84: The reference no. 4 has been provided ([4] Raj P.P. Intervertebral disc: anatomy-physiology-pathophysiology-treatment. Pain Pract. 2008, 8, 18-44)
- Line 121: Please use a more scientific language (not “isn’t so easy”).
Response: These unscientific terms have been removed throughout the whole manuscript.
line 124: What do you mean with “invested costs to get statistically significant results”? Do you mean the costs for the number of animals that have to be used to get meaningful results? Please rephrase.
Response: The reviewer may intend/mean line 105-106: The phrase is replaced by (the costs invested in the experimental groups regarding the required size to get statistically significant results)
- Line 125: There are more reviews that compare animal models (e.g. Alini+ Eur Spine J 2007).
Response: We thank the reviewer. Line 108-109: we cite Alini et al., 2007 now.
- Line 127: Please include references for the different models.
Response: The reviewer may mean line 109: The reference no. 29 has been included. ([35] Daly C.; Ghosh P.; Jenkin G.; Oehme D., Goldschlager T. A Review of Animal Models of Intervertebral Disc Degeneration: Pathophysiology, Regeneration, and Translation to the Clinic. Biomed Res Int. 2016, 2016, 5952165)
Line 135: Please change fetal life to prenatal
Response: The reviewer may mean line 119: it is changed now.
- Line 137: Please remove “directly”
Response: 3.2.: it was removed.
Line 141: Please change “couldn’t” to “could not” contractions should not be used in written English (please change this through the manuscript).
Response: Done. We checked the whole manuscript for this.
- Lines 144-150: Please provide reverences for all methods.
Response: The reviewer may mean line 128-129: References have been provided for each method.
- Line 152: include references for “numerous methods”
Response: It can be found on line 136 now: this phrase “numerous methods” is the own words of the authors, however, the reference no. 39 has been provided ([39] Fajardo R.J.; Karim L.; Calley V.I., Bouxsein M.L. A review of rodent models of type 2 diabetic skeletal fragility. J Bone Miner Res. 2014, 29, 1025-40).
- Line 182: Please spell out Streptozotocin and mention that it is usually used for T1DM. noch erwähnen
Response: Now line 167: Done.
- Line 195: “isn’t a bad model” is not scientific. Please use different word choice. Do you mean that it is a “bad” model? What do you mean with “clear diabetes”? Please be more specific.
Response: The reviewer may intend/mean: “it isn’t a bad candidate” phrase means that it could be employed/used as an acceptable candidate for studying DM, however, this phrase (now line 183-184) has been replaced by “it could be an acceptable candidate”. The phrase “doesn’t develop clear diabetes” means that the model leads to hyperglycemia rather than frank diabetes. The word “clear” has been replaced by “frank”.
- Line 253+255: please remove “huge”, “various” and “finally”, these words are not scientific in this content.
Response: 2.2., line 244: “huge” and “various” have been removed.
Line 246: “finally” has been deleted.
- Line 454: This is the first time AGEs
Response: The reviewer may mean line 444: The abbreviation AGEs is previously mentioned in the line 65 with its description/meaning. Nevertheless, we spelled it once again out now (line 444).
Figures/Figure captions
- Figure 1: A) the bracket for the AF only points to the inner AF, not the outer AF, please mention the degeneration grade and add an image of a non-degenerated human IVD. B1) please use a higher magnification image. B4) The red arrow points to the cartilaginous EP, not the notochordal like cells. Please mention which plane is shown (sagittal, transverse). Orient the images that the caudal site is on the bottom (at the moment the images B4+5 are upside down). Please add scale bars.
Response: We are grateful for the reviewer carefully checking the images for correctness. The figure has been changed accordingly. Unfortunately, we could not provide an image of a healthy unchanged IVD. We describe the grade of degeneration now. Most old donors show IVDD - it is a typical feature of IVD ageing. Hence, we decided to maintain the image providing a nice example of IVDD. We added the planes and scale bars. The images (B4-5) have been correctly turned and mirrored to show the vertebral comlumn in similar orientation. The brackets of AF have been adapted. The red arrow was unfortunately shifted in a wrong position and is corrected now indicating the band with notochordal cells
- Table 3: what does NB mean?
Response: NB has been removed and replaced by “remark” in table 3.
- Table 5: include the first authors (not only the citation) in column 1
Response: Done. We included it also in Table 4.

Reviewer 2 Report
The authors approached to assess potential associations between diabetes mellitus (metabolic disorder) and intervertebral disc disease (musculoscelettal disorder) by reviewing available literature and attempt to provide a narrative review summarizing the relevant findings in a comprehensive way. The topic chosen in this review is of clinical relevance and of great interest, as metabolic disorders of which DM is one of, which have been suspected to impact disorders of the musculuscelettal system (spine, pain). As we observe a rising incidence for both with a considerable burden for the individuals and the society, this review´s topic is highly relevant. Although several studies have been published so far, this review summarizes the available literature considering most importantly the degree of evidence. Saying so, the manuscript is of novel character. The manuscript is well written, despite some minor erros, which can be revised in the proof reading process and enables an easy and informative readibility. The authors conclusion is balanced, in line with the findings of the authors and the authors clearly address the questions posed in their introduction. Overall, I recommend acceptance after minor revision (minor grammar errors)Author Response
Reviewer 2
Comments and Suggestions for Authors
The authors approached to assess potential associations between diabetes mellitus (metabolic disorder) and intervertebral disc disease (musculoscelettal disorder) by reviewing available literature and attempt to provide a narrative review summarizing the relevant findings in a comprehensive way. The topic chosen in this review is of clinical relevance and of great interest, as metabolic disorders of which DM is one of, which have been suspected to impact disorders of the musculuscelettal system (spine, pain). As we observe a rising incidence for both with a considerable burden for the individuals and the society, this review´s topic is highly relevant. Although several studies have been published so far, this review summarizes the available literature considering most importantly the degree of evidence. Saying so, the manuscript is of novel character. The manuscript is well written, despite some minor erros,
which can be revised in the proof reading process and enables an easy and informative readibility.
Response: We checked the manuscript for errors and corrected them.
The authors conclusion is balanced, in line with the findings of the authors and the authors clearly address the questions posed in their introduction. Overall, I recommend acceptance after minor revision (minor grammar errors)
Response: We thank the reviewer for his/her encouraging comments.

Reviewer 3 Report
The manuscript presents a systematic review of the available literature examining the association between type 2 diabetes and intervertebral disc degeneration in rodent models. The interplay between systemic dysregulation and disc degeneration is an area of relatively active research, as such the systematic review of the literature will likely be of interest to disc researchers.
In general, the manuscript would benefit from editing for language and grammar throughout. Many sentences are so poorly structured that the meaning is unclear - for example lines 409-410 “All the studies included irrespectively 
whether performed in vivo or in vitro statistical analyses of data”.
Introduction: There are several statements that are over-generalizations - for example “The NP is located eccentrically (more dorsally) between the central and posterior parts of IVD …”This is not accurate of all IVDs (cervical, thoracic, etc), from animals at all ages, and not consistent between species.
In general the introduction is quite lengthy and superficially covers topics that are not relevant to the review - ie. Large animal models to study disc degeneration.
The Introduction should be more focused. For example, Section 1.4 should be structured based on its title “1.4 Rodents as experimental diabetic models in IVDD” focusing on the strengths, weaknesses and comparison between rodent models (ie rat vs mouse).
Some statements (if they are to be included) should be adequately cited. For example “Recently, genetically modified mice models, were used to investigate the role of specific proteins in the etiology of IVDD pathogenesis [29]. - Surely there is more than one paper that should be cited for a general sentence like this! Moreover, the term recently does not apply since this research has been conducted for decades.
One major issue is specificity of the search - if indeed the search was conducted based on the terms associated with “IVDD in diabetic rat models” as stated in Table 1 - then the title and introduction should be extensively edited to focus on rat and not general rodent studies. It is unclear why the authors would exclude any study of IVDD in diabetic mouse models. This is not well set up in the Introduction or rationalized.
Value of the scoring and assessment of the “Quality of studies included” and evaluation presented as Table 5 is not clear to this reviewer. Much of the scoring system described is based on quantity, not quality, of outcome measures. Such evaluation is particularly biased and may discount very valuable data.
Table 4 should be more succinct. Color coding should be defined in the table legend or footnote
Figures 3 & 4 are redundant and should be revised to clearly present findings in one easy to read figure
Bulleted lists appearing after the study conclusions. These details should be covered within the body of the text in the Discussion in full sentence format. If these points have not been addressed, this section should be revised extensively.
Author Response
Reviewer 3
Comments and Suggestions for Authors
The manuscript presents a systematic review of the available literature examining the association between type 2 diabetes and intervertebral disc degeneration in rodent models. The interplay between systemic dysregulation and disc degeneration is an area of relatively active research, as such the systematic review of the literature will likely be of interest to disc researchers.
In general, the manuscript would benefit from editing for language and grammar throughout.
Response: We edited language and grammar.
Many sentences are so poorly structured that the meaning is unclear - for example lines 409-410 “All the studies included irrespectively 
whether performed in vivo or in vitro statistical analyses of data”.
Response: Line 403: We revised this sentence.
Introduction: There are several statements that are over-generalizations - for example “The NP is located eccentrically (more dorsally) between the central and posterior parts of IVD …” This is not accurate of all IVDs (cervical, thoracic, etc), from animals at all ages, and not consistent between species.
Response: We thank the reviewer for this important criticism and corrected this sequence (see line 24).
In general, the introduction is quite lengthy and superficially covers topics that are not relevant to the review - ie. Large animal models to study disc degeneration.
Response: We shortened the introduction section by removing anatomy of spine and IVDs and removed the statement about large animal models.
The Introduction should be more focused. For example, Section 1.4 should be structured based on its title “1.4 Rodents as experimental diabetic models in IVDD” focusing on the strengths, weaknesses and comparison between rodent models (ie rat vs mouse).
Response: We restructured section 1.4 in regard to a focus on strength and weakness of the models.
Some statements (if they are to be included) should be adequately cited. For example, “Recently, genetically modified mice models, were used to investigate the role of specific proteins in the etiology of IVDD pathogenesis [29]. - Surely there is more than one paper that should be cited for a general sentence like this! Moreover, the term recently does not apply since this research has been conducted for decades.
Response: We support the statement by more references and removed the term: „recently“. Line 127.
One major issue is specificity of the search - if indeed the search was conducted based on the terms associated with “IVDD in diabetic rat models” as stated in Table 1 - then the title and introduction should be extensively edited to focus on rat and not general rodent studies.
Response: We are grateful for the advice. We have included studies, which used rat and mouse models. We focused on both (see title and table 4). We changed “rat” into “rodents” in table 1 and figure 2 since it was a mistake.
It is unclear why the authors would exclude any study of IVDD in diabetic mouse models. This is not well set up in the Introduction or rationalized.
Response: We included already the studies achieved in diabetic mouse models, see table 4, rows 3,6,10,11,12. We have included all the collected related studies, regardless the used diabetic rodent model were perfomed in mouse or rats.
Value of the scoring and assessment of the “Quality of studies included” and evaluation presented as Table 5 is not clear to this reviewer. Much of the scoring system described is based on quantity, not quality, of outcome measures. Such evaluation is particularly biased and may discount very valuable data.
Response: We should differentiate between two different issues: 1) the quality of the study and 2) the strength of evidence, see table 5. The given quality of the included studies was based on a scoring system, which depends on the evaluation of the whole parts (abstract, aim, research question, methods, and results) of each study and the total score of the previously mentioned items gives each study its quality level, while the strength of evidence of each study depends on the total score of the collected evidences. The included proofs must be one of the following determined finding types (statistical, clinical, immunological, imaging, biomechanical, biochemistry, molecular biology or histopathology). Only, the proofs of accepted quality according to the assessment criteria were included in the assessment, while the poor evidences were discarded
Table 4 should be more succinct. Color coding should be defined in the table legend or footnote
Response: We defined the color coding in a footnote of the table 4 now.
Figures 3 & 4 are redundant and should be revised to clearly present findings in one easy to read figure
Response: Figure 3 summarized more global aspects of changes of components of the motion segment (e.g. vertebral bodies, endplates) in addition to the IVD whereas figure 4 focusses on the IVD and the molecular level. We revised both figures and removed redundancies. We refer in Figure 3 to the fact that molecular events in the IVD are depicted in Figure 4.
Bulleted lists appearing after the study conclusions. These details should be covered within the body of the text in the Discussion in full sentence format. If these points have not been addressed, this section should be revised extensively.
Response: We integrated and summarized limitations and strength of this review in full sentences in the discussion section now.

Round 2
Reviewer 1 Report
The authors included most of my comments. Only minor edits are required for publication.
- Figure 1 B2+3: please include that those are transverse cut IVDs
- Table 4: include authors in “Authors” column
- Please change “gender” to “sex” ("Sex" refers to biological differences between females and males... "Gender" refers to socially constructed and enacted roles and behaviors [https://orwh.od.nih.gov/sex-gender])
- The study limitations only need to include the limitations of your study. The other points should go into the discussion: e.g. the second last paragraph (gender differences) should be included in the “gender aspects” sections of the discussion.
Author Response
22st September, 2020
Dear Editor,
On behalf of the authors I would like to thank you for proofreading our manuscript and for the constructive comments.
We modified the manuscript according to your comments and suggestions as explained in our point by point reply listed below. All novel changes performed are in red and underlined in the manuscript.
We hope that our manuscript is up to the Reviewers and Journals expectations now and suitable for publication. Please do not hesitate to contact me for any further questions that may arise concerning the manuscript.
Yours sincerely
Univ.-Prof. Dr. Gundula Schulze-Tanzil
Moderate English changes required
Response: We went through the manuscript and made changes concerning English language where required.
Figure 1 B2+3: please include that those are transverse cut IVDs
Response: We mention now that we showed the transverse plane in the legend of figure 1B2+3.
Table 4: include authors in “Authors” column
Response: Done.
Please change “gender” to “sex” ("Sex" refers to biological differences between females and males... "Gender" refers to socially constructed and enacted roles and behaviors [https://orwh.od.nih.gov/sex-gender])
Response: Done.
The study limitations only need to include the limitations of your study. The other points should go into the discussion: e.g. the second last paragraph (gender differences) should be included in the “gender aspects” sections of the discussion.
Response: We limited the study limitations to our study and the rest was incorporated as suggested in the discussion.
